

**Optimizing sampling strategies in high-resolution paleoclimate records**
Niels J. de Winter[1,2] *, Tobias Agterhuis[1], Martin Ziegler[1]
[1]Department of Earth Sciences, Utrecht University, Princetonlaan 8a, 3584 CB Utrecht, the Netherlands
[2]AMGC research group, Vrije Universiteit Brussel, Pleinlaan 2, 1050 Brussels, Belgium
Correspondence to: Niels J. de Winter (n.j.dewinter@uu.nl)



**Abstract**
The aim of paleoclimate studies to resolve climate variability from noisy proxy records can in essence be
reduced to a statistical problem. The challenge is to isolate meaningful information on climate events from
these records by reducing measurement uncertainty through a combination of proxy data while retaining
the temporal resolution needed to assess the timing and duration of the event. In this study, we explore the
limits of this compromise by testing different methods for combining proxy data (smoothing, binning and
sample size optimization) on a particularly challenging paleoclimate problem: resolving seasonal variability
in stable isotope records. We test and evaluate the effects of changes in the seasonal temperature and
hydrology cycle as well as changes in accretion rate of the archive and parameters such as sampling
resolution and age model uncertainty on the reliability of seasonality reconstructions based on clumped and
oxygen isotope analyses in 33 real and virtual datasets. Our results show that strategic combinations of
clumped isotope analyses can significantly improve the accuracy of seasonality reconstructions if compared
with conventional stable oxygen isotope analyses, especially in settings where the isotopic composition of
the water is poorly constrained. Smoothing data using a moving average often leads to a dampening of the
seasonal cycle, significantly reducing the accuracy of reconstructions. A statistical sample size optimization
protocol yields more precise results than smoothing. However, the most accurate results are obtained
through monthly binning of proxy data, especially in cases where growth rate or water composition cycles
dampen the seasonal temperature cycle. Our analysis of a wide range of natural situations reveals that the
effect of temperature seasonality on isotope records almost invariably exceeds that of changes in water
composition. Thus, in most cases, isotope records allow reliable identification of growth seasonality as a
basis for age modelling and seasonality reconstructions in absence of independent chronological markers
in the record. These specific findings allow us to formulate general recommendations for sampling and
combining data in paleoclimate research and have implications beyond the reconstruction of seasonality.
We discuss the implications of our results for solving common problems in paleoclimatology and
stratigraphy, including cyclostratigraphy, strontium isotope dating and event stratigraphy.



## 1. Introduction

Improving the resolution of climate reconstructions is a key objective in paleoclimate studies because it allows climate variability to be studied on different timescales and sheds light on the continuum of climate variability (Huybers and Curry, 2006). However, the temporal resolution of climate records is limited by the accretion rate (growth or sedimentation rate) of the archive and the spatial resolution of sampling for climate reconstructions, which is a function of the size of samples required for a given climate proxy. This tradeoff between sample size and sampling resolution is especially prevalent when using state-of-the-art climate proxies which require large sample sizes, such as the carbonate clumped isotope paleothermometer ($\Delta_{47}$; see applications in Rodríguez-Sanz et al., 2017; Briard et al., 2020) or stable isotope ratios in specific compounds or of rare isotopes (e.g. phosphate-oxygen isotopes in tooth apatite, triple oxygen isotopes in speleothems or carbon isotopes of $CO_2$ in ice cores; Jones et al., 1999; Schmitt et al., 2012; Sha et al., 2020). The challenge of sampling resolution persist on a wide range of timescales: from attempts to resolve geologically short-lived (kyr-scale) climate events from deep sea cores with low sedimentation rates (e.g. Stap et al., 2010; Rodríguez-Sanz et al., 2017) to efforts to characterize tidal or daily variability in accretionary carbonate archives (e.g. Warter and Müller, 2017; de Winter et al., 2020a). What constitutes "high-resolution" is therefore largely dependent on the specifics of the climate archive.

Sample size limitations are especially important in paleoseasonality reconstructions. Reliable archives for seasonality (e.g. corals, mollusks and speleothem records) are in high demand in the paleoclimate community, because the seasonal cycle is the most important cycle in Earth's climate and seasonality reconstructions complement more common long-term (kyr to -Myr) records of past climate variability (e.g. Morgan and van Ommen, 1997; Tudhope et al., 2001; Steuber et al., 2005; Steffensen et al., 2008; Denton et al., 2005; Huyghe et al., 2015; Vansteenberge et al., 2019). A more detailed understanding of climate dynamics at the human timescale is increasingly relevant for improving climate projections (IPCC, 2013). Unfortunately, the growth and mineralization rates of archives that capture high-resolution variability (rarely exceeding 10 mm/yr) limit the number and size of samples that can be obtained at high temporal resolutions (e.g. Mosley-Thompson et al., 1993; Passey and Cerling, 2002; Treble et al., 2003; Goodwin et al., 2003).



A promising technique for circumventing sample size limitations is to analyze larger numbers of small
aliquots from the same sample or from similar parts of the climate archive. These smaller aliquots typically
have a poorer precision, but averaging multiple aliquots into one estimate while propagating the
measurement uncertainty leads to a more reliable estimate of the climate variable (Dattalo, 2008; Meckler
et al., 2014; Müller et al., 2017; Fernandez et al., 2017). This approach yields improved sampling flexibility
since aliquots can be combined in various ways after measurement. It also allows outlier detection at the
level of individual aliquots, thereby spreading the risk of instrumental failure and providing improved control
on changes in measurement conditions that may bias results.
Previous studies have applied several different methods for combining data from paleoclimate records to
reduce analytical noise or higher order variability, and extract variability with a specific frequency (e.g. a
specific orbital cycle or seasonality; e.g. Lisiecki and Raymo, 2004; Cramer et al., 2009). These data
reduction approaches can in general be categorized into: **smoothing** techniques, in which a sliding window
or range of neighboring datapoints is used to smooth high resolution records (see e.g. Cramer et al., 2009)
or **binning** techniques, in which the record is divided into equal bins along its length axis (e.g. time, depth
or length in growth direction; e.g. Lisiecki and Raymo, 2004). In addition, a third approach is proposed here
based on **optimization** of sample size for dynamic binning of data along the climate cycle using a moving
window in the domain of the climate variable (as opposed to the depth domain) combined with a T-test
routine (see  section 3.**4**). All three approaches have advantages and caveats.

**2. Aim**
In this study, we explore the (dis)advantages of these three data reduction approaches by testing their
reliability in resolving seasonal variability in sea surface temperature (SST) and seawater stable oxygen
isotope composition ($\delta^{18}O_{sw}$), both highly sought-after variables in paleoclimate research. We compare
reconstructions of SST and $\delta^{18}O_{sw}$ in real and virtual datasets from accretionary carbonate archives (e.g.
shells, corals and speleothems) using the clumped isotope thermometer ($\Delta_{47}$) combined with stable oxygen
isotope ratios of the carbonate ($\delta^{18}O_c$). Throughout the remainder of this work, the three methods for
combining data for reconstructions are abbreviated as follows (see also **Fig. 1** and 3.**4**):





**Smoothing:** Reconstructions of SST and $\delta^{18}O_{sw}$ based on **moving averages** of $\Delta_{47}$ records.
**Binning**: Reconstructions of SST and $\delta^{18}O_{sw}$ based on binning of $\Delta_{47}$ records into **monthly time bins**.
**Optimization** Reconstructions of SST and $\delta^{18}O_{sw}$ based on **sample size optimization** in $\Delta_{47}$ records.
For comparison, we also include reconstructions based purely on individual $\delta^{18}O_c$ measurements with an
(often inaccurate) assumption of constant $\delta^{18}O_{sw}$, which form the most common method for carbonate-
based temperature reconstructions in paleoclimate research. These reconstructions were not subject to
any of the data combination methods outlined above and mostly serve as a benchmark to compare with
the performance of the $\Delta_{47}$ methods. SST reconstructions assuming constant $\delta^{18}O_{sw}$ are hereafter referred
to as "**$\delta^{18}O$**" reconstructions.
We evaluate the reliability of all four approaches through measures of accuracy (offset of reconstruction
from the true value) and precision (variability between reconstructions due to random errors in the data) of
reconstructions and highlight biases inherent to specific approaches and in specific situations. In the end,
we provide guidelines for choosing the right sampling approach for studies on seasonality reconstructions
from accretionary carbonate archives. In addition, we discuss implications of our findings for other sampling
problems in the geosciences.



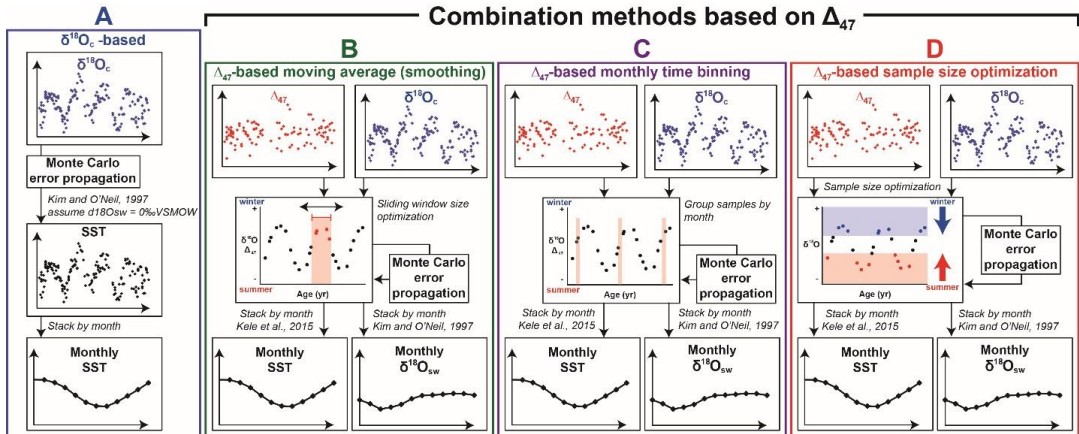


**Figure 1**: *Schematic overview of the four approaches for seasonality reconstructions: (**A**) δ$^{18}$O-based reconstructions, assuming constant δ$^{18}$O$_{sw}$. (**B**) Reconstructions based on **smoothing** δ$^{18}$O$_c$ and Δ$_{47}$ data using a moving average. (**C**) Reconstructions based on binning δ$^{18}$O$_c$ and Δ$_{47}$ data in monthly time bins. (**D**) Reconstructions based on **optimization** of the sample size for combining δ$^{18}$O$_c$ and Δ$_{47}$ data (see description in 3.4). Colored curves represent virtual δ$^{18}$O$_c$ (blue) and Δ$_{47}$ (red) depth series. Black curves represent reconstructed monthly SST and δ$^{18}$O$_{sw}$ averages.*

**3. Methods**

**3.1 SST and δ$^{18}$O$_{sw}$ data**

The reliability (accuracy and precision) of approaches was illustrated and tested in three ways: Firstly, by

evaluating data from a real specimen of a Pacific oyster (*Crassostrea gigas*, syn. *Magallana gigas*) reported

in Ullmann et al. (2010). Secondly, by application on data based on actual measurements of natural

variability in SST and sea surface salinity (SSS; case 30-33). Thirdly, by applying the approaches on set of

virtual datasets based on completely virtual SST and δ$^{18}$O$_{sw}$ data (case 1-29). For virtual datasets, records

of SST and δ$^{18}$O$_{sw}$ were converted to the depth domain (along the length of the record) by defining a virtual

growth rate in the sampling direction. Adding this growth rate as a variable allowed us to test the sensitivity

of approaches to changes in the extension rate of the archive, including hiatuses (growth rate = 0). This is

important, because fluctuations in linear extension rate and periods in which no mineralization occurs

(hiatuses or growth cessations) are common in all climate archives (e.g. Treble et al., 2003; Ivany, 2012).

An overview of the virtual SST and δ$^{18}$O$_{sw}$ time series in all test cases is shown in **Fig. 2** and a description

of all cases is given in **S1**.

**Virtual cases**

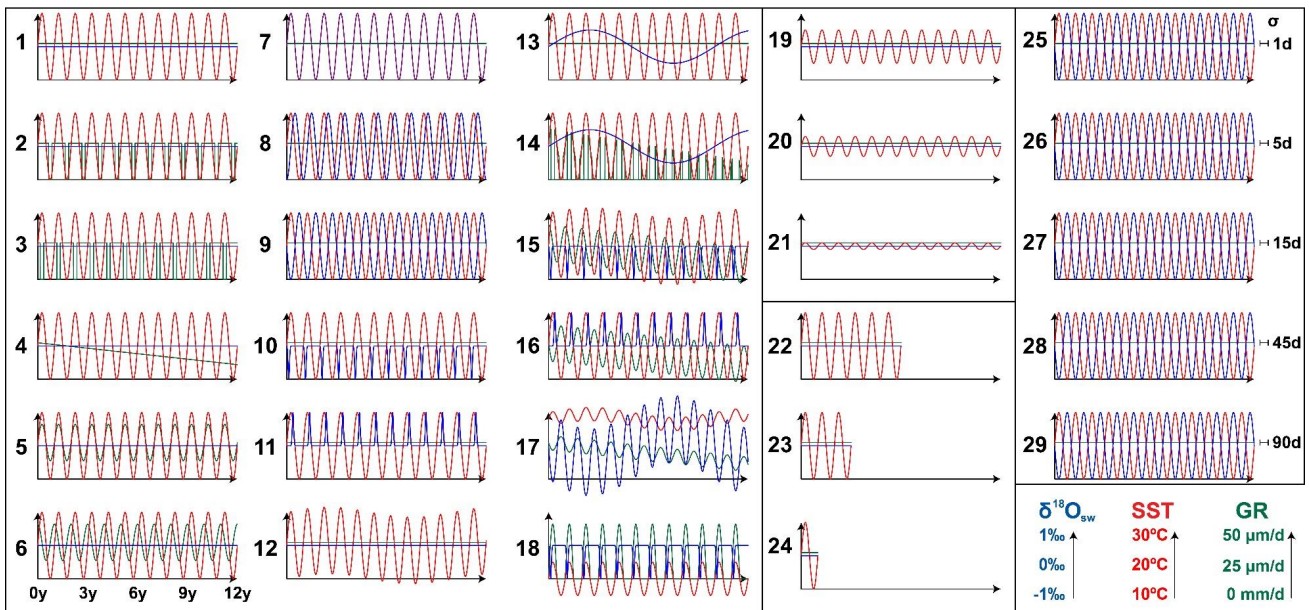

Figure 2: Overview of time series of all virtual test cases. Colored curves represent time series of SST (red), $\delta^{18}O_{sw}$ (blue) and growth rate (green, abbreviated as "GR"). Horizontal axes in all plots are 12 years long (see legend below case 6). Vertical axis of all plots has the same scale (SST: 10 to 30°C; $\delta^{18}O_{sw}$: -1 to +1‰ VSMOW; Growth rate: 0 –50 μm/day; see legend in bottom right corner). Horizontal error bars and labels on the right side of cases 25-29 represent standard errors introduced on the age model (bars not to scale). The $\delta^{18}O_c$ and $\Delta_{47}$ records resulting from these virtual datasets are provided in S8 (see also Fig. 3 for natural examples).





*2.1.1 Modern oyster data*
Environmental SST and $\delta^{18}O_{sw}$ data from the List Basin in Denmark (54°59.25N, 8°23.51E) where the
modern oyster specimen originated were obtained from local *in situ* measurements of SST and SSS
described in Ullmann et al. (2010). Since direct, *in situ* measurements of $\delta^{18}O_{sw}$ variability at a high temporal
resolution were not available, $\delta^{18}O_{sw}$ was estimated from (more widely available) SSS data using a mass
balance (equation 1 and 2; following e.g. Ullmann et al., 2010):
$$\delta^{18}O_{sw} = \delta^{18}O_{sw,freshwater} * f + \delta^{18}O_{sw,freshwater} * (1 - f) \qquad (1)$$
$$f = \frac{SSS_{sample} - SSS_{ocean}}{SSS_{freshwater} - SSS_{ocean}} \qquad (2)$$
Here, we assume salinity ($SSS_{sample}$) results from a mixture of a fraction ($f$) isotopically light and low-salinity
($\delta^{18}O_{sw,freshwater}$ = -8.5‰VSMOW; $SSS_{freshwater}$ = 0 ) freshwater and a fraction ( *1-f*) ocean water ($\delta^{18}O_{sw,ocean}$
= 0‰VSMOW; $SSS_{ocean}$ = 35 ), with negative amounts of freshwater contribution ( $f$ < 0) representing net
evaporation ($SSS_{sample}$ > $SSS_{ocean}$). The value for $\delta^{18}O_{sw,freshwater}$ was based on the discharge weighted
average $\delta^{18}O_{sw}$ of water in the nearby Elbe and Weser rivers (-8.5‰VSMOW; see Ullmann et al., 2010).
*3.1.2 Cases based on real climate data*
Natural environmental time series were based on SST and SSS data from four different locations, selected
to capture a variety of environments with different SST and SSS variability:
1.  Tidal flats of the Wadden Sea near Texel, the Netherlands (case 30)
2.  Great Barrier Reef in Australia (case 31)
3.  Gulf of Aqaba between Egypt and Saudi Arabia (case 32)
4.  Northern Atlantic Ocean east of Iceland (case 33).
Daily measurements of SST and SSS for case 31-33 were obtained from worldwide open-access datasets
of the National Oceanic and Atmospheric Administration (NOAA, 2020) and European Space Agency (ESA,
2020) respectively. Hourly SST and SSS measured *in situ* in the Wadden Sea (case 30) were obtained
from the Dutch Institute for Sea Research (NIOZ, Texel, the Netherlands). Since direct, *in situ*
measurements of $\delta^{18}O_{sw}$ variability at a high temporal resolution is scarce, $\delta^{18}O_{sw}$ was estimated from (more





widely available) SSS data using the same mass balance described in **3.1.1**. The value for $\delta^{18}O_{sw,freshwater}$
was based on the $\delta^{18}O_{sw}$ of rain in the Netherlands (-8‰VSMOW; Mook, 1970; Bowen, 2020), and applying
this mass balance on the SSS record of the Wadden Sea tidal flats (case 30) results in $\delta^{18}O_{sw}$ values and
a SSS-$\delta^{18}O_{sw}$ relationship in agreement with measurements in this region (Harwood et al., 2008). SST and
$\delta^{18}O_{sw}$ time series for all cases are given in **S5** and natural cases are plotted in **Fig. 3**.



# Real data based cases



**Figure 3**: *Overview of the four cases of virtual data based on natural SST and SSS measurements explored in this study. (**A**) Case 30: Tidal flats on the Wadden Sea, Texel, the Netherlands. (**B**) Case 31 Great Barrier Reef, Australia). (**C**) Case 32: Gulf of Aqaba between Egypt and Saudi Arabia. (**D**) Case 33: Atlantic Ocean east of Iceland. For all cases, graphs on top show environmental data, with SST plotted in red, $\delta^{18}O_{sw}$ in blue and growth rate (abbreviated as "GR") in green (as in **Fig. 2**). The graph below shows virtual $\delta^{18}O_c$ (blue) and $\Delta_{47}$ (red) records created from these data series using a sampling interval of 0.45 mm and including analytical noise (see 3.**3**). Note that the scale of vertical axes varies between plots.*



*2.1.3 Virtual cases*
Virtual SST and $\delta^{18}O_{sw}$ time series were constructed to test the effect of various SST and $\delta^{18}O_{sw}$ scenarios
on the effectivity of the reconstruction methods. The default test case (case 1) contained an ideal, 12-year
sinusoidal SST curve with a period of 1 year (seasonality), a mean value of 20°C and a seasonal amplitude
of 10°C, a constant $\delta^{18}O_{sw}$ value of 0‰VSMOW and a constant growth rate of 10 mm/yr. Other cases
contain various deviations from this ideal case (see also **S1**):

•   Linear and/or seasonal changes in growth rate, including growth stops (cases 2-6, 14-18)

•   Seasonal and/or multi-annual changes in $\delta^{18}O_{sw}$ (cases 7-11, 13-18)

•   Multi-annual trends in SST superimposed on the seasonality (cases 12, 15 and 17)

•   Variations in the seasonal SST amplitude (cases 19-21)

•   Change in the total length of the time series (cases 22-24).

•   Variation in uncertainty on the age of each virtual datapoint (cases 25-29)

Comparison of the virtual time series (case 1-29; **Fig. 2**) with the natural variability (case 30-33; **Fig. 3**)
shows that the virtual cases are not realistic approximations of natural variability in SST and $\delta^{18}O_{sw}$. Natural
SST and $\delta^{18}O_{sw}$ variability are not limited to the seasonal or multi-annual scale but contain a fair amount of
higher order (daily to weekly scale) variability. In order to simulate this natural variability, we extracted the
seasonal component of SST and $\delta^{18}O_{sw}$ variability from our highest resolution record of measured natural
SST and SSS data (case 30: data from Texel, the Netherlands, see **3.1.2** and **Fig. 3**). The standard
deviation of residual variability of this data after subtraction of the seasonal cycle was used to add random
high-frequency noise to the SST and $\delta^{18}O_{sw}$ variability in virtual cases. Note that while sub-annual
environmental variability can be approximated by Gaussian noise (Wilkinson and Ivany, 2002), this
representation is an oversimplification of reality. In the case of our Texel data, the SST and SSS residuals
are not exactly normally distributed (Kolmogorov-Smirnov test: $D = 0.010$; $p = 7.2*10^{-14}$ and $D = 0.039$; $p <$
$2.2*10^{-16}$ for SST and SSS residuals respectively; see **S2-4**).





### 3.2. Subsampling

Virtual aliquots were subsampled at equal distance from the SST and $\delta^{18}O_{sw}$ depth series of all cases using
six sampling intervals: 0.1 mm, 0.2 mm, 0.45 mm, 0.75 mm, 1.55 mm and 3.25 mm. The four largest
sampling intervals were chosen such that the standard growth rate (10 mm/yr) was not an integer multiple
of the sampling interval (e.g. 0.45 mm instead of 0.5 mm, and 3.25 mm instead of 3 mm). This decision
prevents sampling the same parts of the seasonal cycle (e.g. same months) every year, which biases both
the mean value and the precision of monthly SST and $\delta^{18}O_{sw}$ reconstructions. This bias towards certain
parts of the seasonal cycle is much stronger at low sample sizes (large sampling intervals) and is illustrated
in **S6**.

### 3.3 Conversion to $\delta^{18}O_c$ and $\Delta_{47}$

After subsampling, SST and $\delta^{18}O_{sw}$ were converted to $\delta^{18}O_c$ and $\Delta_{47}$ using a carbonate model based on
empirical relationships of $\Delta_{47}$ and $\delta^{18}O_c$ with and SST and $\delta^{18}O_{sw}$ (equation 3 and 4; Kim and O'Neil, 1997;
Kele et al., 2015; Bernasconi et al., 2018) and the conversion of $\delta^{18}O$ values from VSMOW to VPDB scale
(equation 5; Brand et al., 2014):
$$\Delta_{47} = \frac{0.0449 * 10^6}{(SST + 273.15)^2} + 0.167 \qquad (3)$$
$$1000 * \ln \frac{\left(^{18}O/_{16}O\right)_{CaCO_3}}{\left(^{18}O/_{16}O\right)_{H_2O}} = 18.03 * \left(\frac{10^3}{(SST + 273.15)}\right) - 32.42 \qquad (4)$$
$$\delta^{18}O_{VPDB} = 0.97002 * \delta^{18}O_{VSMOW} - 29.98 \qquad (5)$$
The resulting depth records of $\Delta_{47}$ and $\delta^{18}O_c$ and their associated true SST and $\delta^{18}O_{sw}$ records are used as
basis for comparing the reliability of the approaches in different scenarios. A schematic overview of all steps
taken to create virtual data and test the four reconstruction approaches as well as an example of case 30
(Great Barrier reef data, see also **Fig. 2**) is provided in **Fig. 4**. All calculations for creating $\Delta_{47}$ and $\delta^{18}O_c$
depth series were carried out using the open-source computational software R (R core team, 2013), and
scripts for these calculations are given in **S7**. All $\Delta_{47}$ and $\delta^{18}O_c$ datasets are provided in **S8**. In the case of
the real oyster data, $\delta^{18}O_c$ data from Ullmann et al. (2010) was used and $\Delta_{47}$ data was created from the





seasonal SST record provided in the same study with added natural residual variability (as explained in
**3.1.3**).






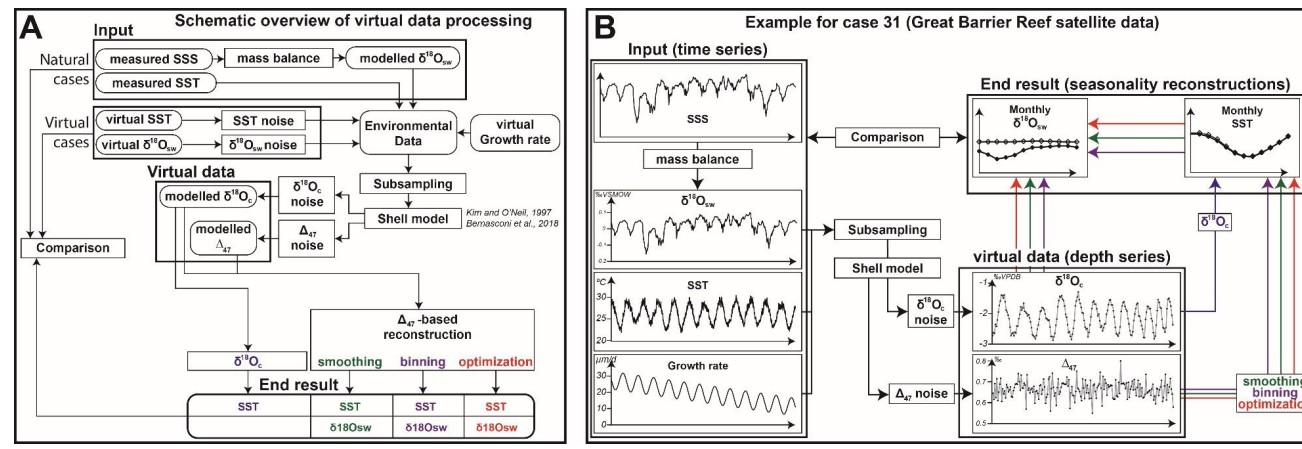

**Figure 4**: **A**) Flow diagram showing the steps taken to create virtual data and compare results of SST and δ¹⁸O$_{sw}$ reconstructions with the actual
SST and δ¹⁸O$_{sw}$ data the record was based on. **B**) An example of the steps highlighted in **A**) using case 31 (Great Barrier Reef data) meant to
illustrate the data processing steps. Virtual data plots include normally distributed measurement uncertainty on Δ$_{47}$ and δ¹⁸O$_c$



### 3.4 SST and δ¹⁸O$_c$ reconstructions

SST and δ¹⁸O$_{sw}$ seasonality were reconstructed from the Δ$_{47}$ and δ¹⁸O$_c$ records to test the reliability of the sample reduction approaches (see **Fig. 1**). In all approaches, a typical analytical uncertainty on measurements of Δ$_{47}$ (one standard deviation of 0.04‰) and δ¹⁸O$_c$ (one standard deviation of 0.05‰) was used to include measurement precision. These analytical uncertainties were chosen based on typical uncertainties reported for these measurements in the literature (e.g. Schöne et al., 2005; Huyghe et al., 2015; Vansteenberge et al., 2016) and long-term precision uncertainties obtained by measuring in-house standards using the MAT253+ with Kiel IV setup in the clumped isotope laboratory at Utrecht University (e.g. Kocken et al., 2019). Virtual measurement uncertainty was propagated through all reconstruction approaches using a Monte Carlo simulation (N = 1000) in which Δ$_{47}$ and δ¹⁸O$_c$ records were randomly sampled from a normal distribution with the virtual Δ$_{47}$ and δ¹⁸O$_c$ values as means and analytical uncertainties as standard deviations. For each case study, sampling interval and reconstruction method, SST and δ¹⁸O$_{sw}$ results were aggregated into monthly averages, medians, standard deviations, and standard errors. Step by step documentation of calculations made for the three Δ$_{47}$-based reconstruction approaches and the δ¹⁸O$_c$ reconstructions are given in **S9** and are detailed below.

For **δ¹⁸O** reconstructions (**Fig. 1A**), only the δ¹⁸O$_c$ records were used. Seawater δ¹⁸O$_{sw}$ values were assumed to remain 0‰VSMOW throughout the year. The simulated δ¹⁸O$_c$ records with analytical uncertainties added were directly converted to SST using the Kim and O'Neil (1997) temperature relationship (see equation 4).

**Smoothing** reconstructions (**Fig. 1B**) were carried out by defining a range of moving window sizes (from N=1 to the complete record). For every simulated Δ$_{47}$ and δ¹⁸O$_c$ record, all moving windows were tested. The window size that resulted in the most significant difference between maximum and minimum Δ$_{47}$ values using a student's T-test was applied on both Δ$_{47}$ and δ¹⁸O$_c$ records. This process was repeated for all virtual records to propagate simulated analytical uncertainty through the protocol. SST and δ¹⁸O$_{sw}$ were calculated for each set of Δ$_{47}$ and δ¹⁸O$_c$ records using the combination of empirical temperature relationships by Kim and O'Neil (1997) and Bernasconi et al. (2018; equation 3)



In **binning** reconstructions (**Fig. 1C**), virtual $\Delta_{47}$ and $\delta^{18}O_c$ data were grouped into monthly time bins and
converted to SST and $\delta^{18}O_{sw}$ using the Kim and O'Neil (1997) and Bernasconi et al. (2018) formulae. The
prerequisite for this method is that the data is aligned using a (floating) age model accurate enough to allow
samples to be placed in the right bin. The age of virtual samples in this study is known so this prerequisite
poses no problems in this case, but the same may not be true in the fossil record.
Finally, the **optimization** reconstruction approach (**Fig. 1D**) was carried out by ordering the aliquots of each
virtual dataset from warm (low $\delta^{18}O_c$) to cold (high $\delta^{18}O_c$ data) samples, regardless of their position relative
to the seasonal cycle. From this ordered dataset, increasingly large samples of multiple aliquots (from N=1
to the complete record) are taken from both the warm ("summer") and the cold ("winter") side of the
distribution. Sample sizes with significant difference in $\Delta_{47}$ value between summer and winter groups (p ≤
0.05 based on a student's T-test) were selected as optimal sample sizes. For each successful sample size,
SST and $\delta^{18}O_{sw}$ values were calculated from $\Delta_{47}$ and $\delta^{18}O_c$ data according to Kim and O'Neil (1997) and
Bernasconi et al. (2018) formulae. The relationship between SST and $\delta^{18}O_{sw}$ obtained from these
reconstructions was used to convert all data to SST and $\delta^{18}O_{sw}$.
Accuracy and precision of reconstructions of the following four parameters were evaluated:

1.  mean annual SST (MAT)

2.  seasonal range in SST (temperature difference between warmest and coldest month)

3.  mean annual $\delta^{18}O_{sw}$

4.  seasonal range in $\delta^{18}O_{sw}$ ($\delta^{18}O_{sw}$ difference between warmest and coldest month).

Accuracy was defined as the absolute offset of the reconstruction from the actual data. Precision was
defined as the (relative) standard deviation of the reconstruction, as calculated from the variability within
monthly time bins resulting from error propagation through the reconstruction methods. An overview of
monthly SST and $\delta^{18}O_{sw}$ reconstructions using the four approaches in all cases is given in **S5**. Raw data
results and figures of reconstructions of all cases using all sampling resolutions are compiled in **S10**.






## 4. Results

### 4.1 Real example

Measured ($\delta^{18}O_c$) and simulated ($\Delta_{47}$) data from the Pacific oyster from the Danish List Basin yield various estimates for SST and $\delta^{18}O_{sw}$ seasonality depending on which reconstruction approach is taken (**Fig. 5**). While a model of shell $\delta^{18}O_c$ based on SST and SSS data closely approximates the measured $\delta^{18}O_c$ record (**Fig. 5C**), basing SST reconstructions solely on $\delta^{18}O_c$ data without any *a priori* knowledge of $\delta^{18}O_{sw}$ variability (assuming constant $\delta^{18}O_{sw}$ equal to the global marine value) leads to high inaccuracy in SST seasonality and mean annual SST (**Fig. 5D**). The in-phase relationship between SST and SSS (**Fig. 5B**) dampens the seasonal $\delta^{18}O_c$ cycle, causing underestimation of temperature seasonality, while a negative mean annual $\delta^{18}O_{sw}$ value in the List Basin biases SST reconstructions towards higher temperatures. In terms of SST reconstructions, the **smoothing**, **binning** and **optimization** approaches based on $\Delta_{47}$ and $\delta^{18}O_c$ data yield more accurate reconstructions, albeit with a reduced seasonality and a bias towards the summer season. The latter is a result of severely reduced growth rates in the winter season, which was therefore undersampled (see **Fig. 5A** and **5C**). Approaches including $\Delta_{47}$ data also yield far more accurate $\delta^{18}O_{sw}$ estimates than the **$\delta^{18}O$** approach. However, the accuracy on both seasonality and mean annual $\delta^{18}O_{sw}$ estimates is high in these approaches too, largely because of the limited sampling resolution, especially in winter. The **optimization** approach suffers especially from the strong in-phase relationship between SST and SSS, which obscures the difference between the $\delta^{18}O_{sw}$ effect and the temperature effect on shell carbonate. Yet, disentangling SST from $\delta^{18}O_{sw}$ seasonality is central to the success of the approach (see **3.4**). **Fig. 5D** does not show the reproducibility error on SST and $\delta^{18}O_{sw}$ estimates, which is much larger for the **smoothing** approach than for the **binning** an **optimization** approaches due to the limited data in the winter seasons (see **S5**).

These results highlight that several properties of carbonate archives, such as growth rate variability, phase relationships between SST and $\delta^{18}O_{sw}$ seasonality and sampling resolution, can negatively impact the reliability of paleoseasonality reconstructions. The virtual and real data cases in this study were tailored to test the effects of these archive properties more thoroughly.

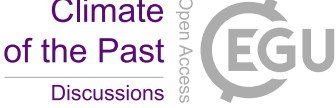

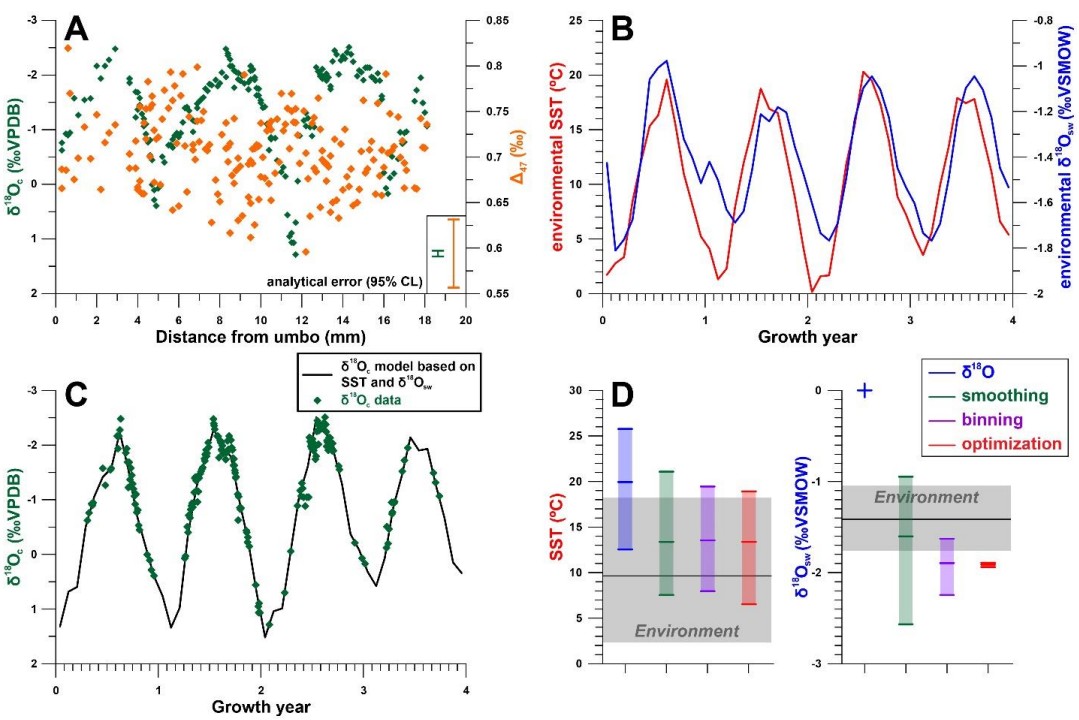

**Figure 5**: (**A**) Plot of $\delta^{18}O_c$ and (virtual) $\Delta_{47}$ data from a modern Pacific oyster (Crassostrea gigas; see
Ullmann et al., 2010). (**B**) shows SST and $\delta^{18}O_{sw}$ data from the List Basin (Denmark) in which the oyster
grew. (**C**) shows the fit between $\delta^{18}O_c$ data and modelled $\delta^{18}O_c$ calculated from SST and $\delta^{18}O_{sw}$ on which
the shell age model was based. (**D**) Shows a summary of the results of different approaches for
reconstructing SST and $\delta^{18}O_{sw}$ from the $\delta^{18}O_c$ and $\Delta_{47}$ data. The vertical colored bars show the
reconstructed seasonal variability using all methods with ticks indicating warmest month, coldest month,
and annual mean. The grey horizontal bars show the actual seasonal variability in the environment.
Precision errors on monthly reconstructions are not shown but are given in **S5**.


**3.2 Case specific results**
A case-by-case breakdown of the precision (**Fig. 6**) and accuracy (**Fig. 7**) of reconstructions using the four
approaches shows that reliability of reconstructions varies significantly between approaches and is highly
case-specific. In general, precision is highest in **δ¹⁸O** reconstructions, followed by **optimization** and
**binning** with **smoothing** generally yielding the worst precision. Average precision standard deviations of
the underperforming methods (**binning** and **smoothing**) are up to 2-3 times larger than those of **δ¹⁸O** (e.g.
respectively 3.9°C and 3.5°C vs. 1.3°C for **δ¹⁸O** MAT reconstructions). It is worth noting that precision on
**δ¹⁸O**-based estimates is mainly driven by measurement precision (which is better for $\delta^{18}O$ than for $\Delta_{47}$



measurements, see section **5.1.1**), while $\Delta_{47}$-based reconstructions lose precision due to the higher
measurement error on $\Delta_{47}$ measurements and the method used for combining measurements for
seasonality reconstructions. On a case-by-case basis, the hierarchy of approaches can differ, especially if
strong variability in growth rate is introduced, such as in case 14, where the size of hiatuses in the record
increases progressively, or in case 18, in which half of the year is missing due to growth hiatuses (see **S1**
and **S5**). Between the $\Delta_{47}$-based methods (**smoothing**, **binning** and **optimization**), **optimization** is rarely
outcompeted in terms of precision in both SST and $\delta^{18}O_{sw}$ reconstructions.
The comparison based on precision alone is misleading, as the approach which is most precise (**$\delta^{18}O$**) runs
the risk of being highly inaccurate (offsets exceeding 4°C on some MAT reconstructions; see **Fig. 7C**),
especially in cases based on natural SST and SSS (case 30-33). The **smoothing** approach also often
yields highly inaccurate results, especially in cases with substantial variability in $\delta^{18}O_{sw}$ (e.g. case 9-11).
Accuracy of **optimization** and **binning** outcompete the other methods in most circumstances. **Binning**
outperforms **optimization** in reconstructions of $\delta^{18}O_{sw}$ seasonality, making it overall the most accurate
approach. Interestingly, **optimization** is less accurate specifically in cases with sharp changes in growth
rate in summer (e.g. cases 11, 14, 16 and 17), with **binning** performing better in these cases.
Reconstructions of mean annual SST and $\delta^{18}O_{sw}$ of case 18 are especially inaccurate regardless of which
method is applied. This extreme case with hiatuses lasting half of the year combined with seasonal
fluctuations in both SST and $\delta^{18}O_{sw}$ presents a worst-case scenario for seasonality reconstructions leading
to strong biases in mean annual temperature reconstructions. In situations like case 18, the **optimization**
approach is most accurate in MAT and SST seasonality reconstructions, but $\delta^{18}O_{sw}$ is more accurately
reconstructed using the **binning** approach. Finally, it is worth noting that in natural situations (**Fig. 3**),
variability in SST almost invariably has a larger influence on $\delta^{18}O$ and $\Delta_{47}$ records, such that fluctuations in
$\delta^{18}O_c$ records closely follow the SST seasonality even in cases with relatively large $\delta^{18}O_{sw}$ variability (e.g.
case 30). Chronologies based on these $\delta^{18}O$ fluctuations are therefore generally accurate.



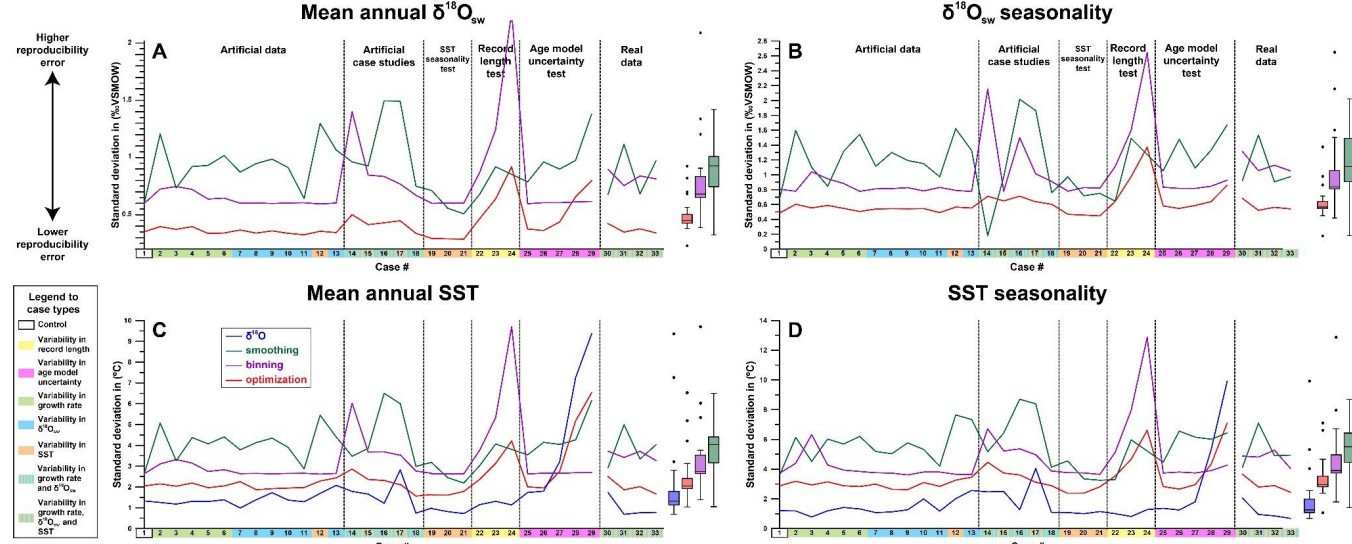

**Figure 6**: *Overview of precision (one standard deviation) of reconstructions of mean annual δ¹⁸O$_{sw}$ (**A**), seasonal range in δ¹⁸O$_{sw}$ (**B**), mean annual SST (**C**) and seasonal range in SST (**D**), with higher values indicating lower precision (higher precision errors) based on average sampling resolution (sampling interval of 0.45 mm). The horizontal axis displays the different cases, color coded by their difference from the control case (case 1; see legend on the left-hand side). Colored lines indicate the different data treatment approaches. Box-whisker plots to the right show medians and distributions of precision on cases using different reconstruction approaches (outliers are identified as black dots based on 2x interquartile distance). Color coding follows the scheme in **Fig. 1**.*





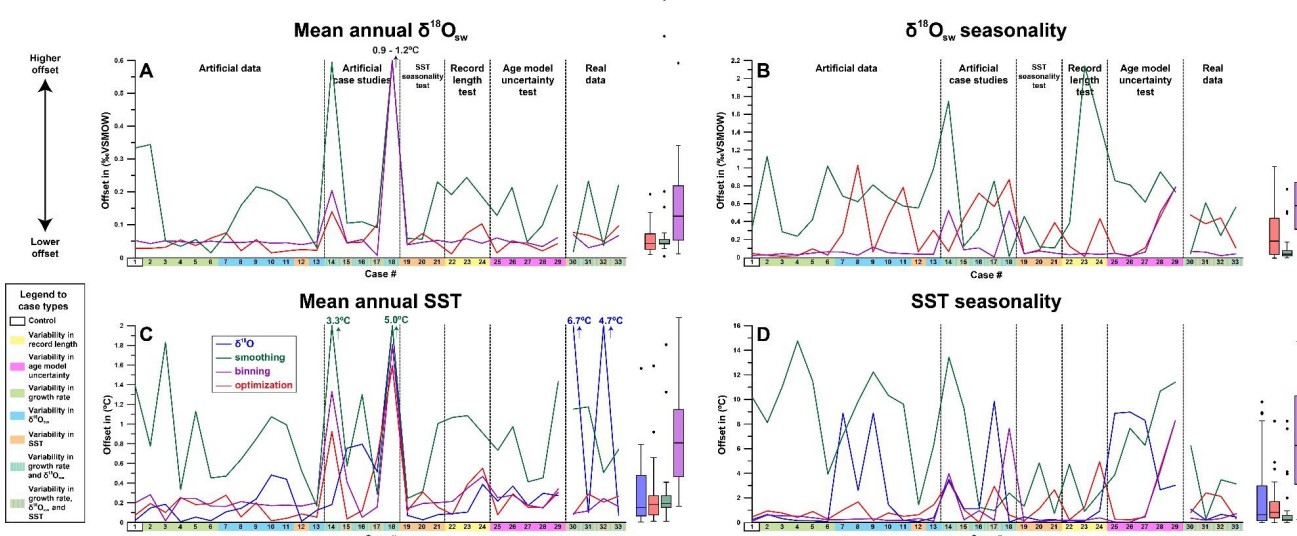

**Figure 7**: *Overview of accuracy (absolute offset from actual values) of reconstructions of mean annual $\delta^{18}O_{sw}$ (**A**), seasonal range in $\delta^{18}O_{sw}$ (**B**), mean annual SST (**C**) and seasonal range in SST (**D**), with higher values indicating lower accuracy (higher offsets) based on average sampling resolution (sampling interval of 0.45 mm). The horizontal axis displays the different cases, color coded by their difference from the control case (case 1; see legend on the left-hand side). Box-whisker plots to the right show medians and distributions of accuracy on cases using different reconstruction approaches (outliers are identified as black dots based on 2x interquartile distance). Color coding follows the scheme in **Fig. 1** and **Fig. 6**.*



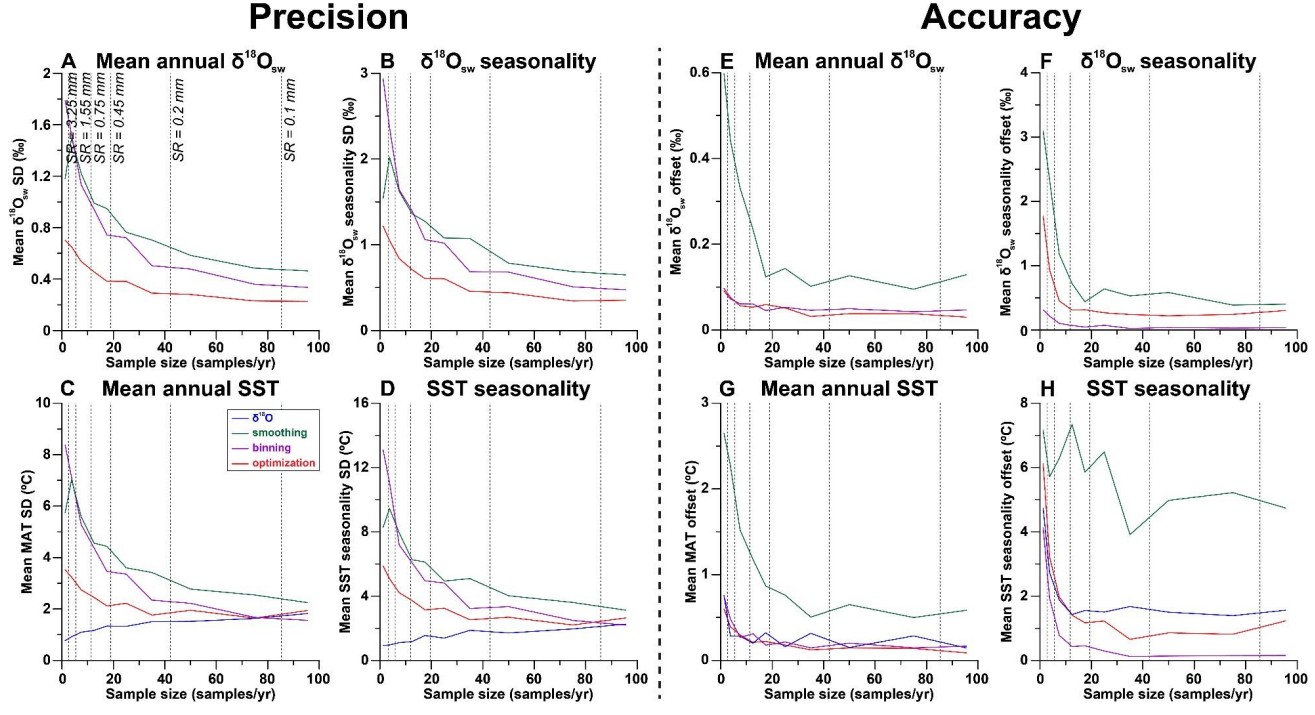

Figure 8: Effect of sampling resolution (in samples per year, see **S5**) on the precision (one standard deviation) of results of reconstructions of mean annual $\delta^{18}O_{sw}$ (**A**), seasonal range in $\delta^{18}O_{sw}$ (**B**), mean annual SST (**C**) and seasonal range in SST (**D**). Effect on the accuracy (absolute offset from actual value) of results of reconstructions of mean annual $\delta^{18}O_{sw}$ (**E**) and seasonal range in $\delta^{18}O_{sw}$ (**F**), mean annual SST (**G**) and seasonal range in SST (**H**). Color coding follows the scheme in **Fig. 1** and **Fig. 4**.





**4.3 Effect of sampling resolution**
As expected, increasing the temporal sampling resolution (i.e. number of samples per year) almost
invariably increases the precision and accuracy (**Fig. 8**) of reconstructions using all methods. An exception
to this rule is the precision of $\delta^{18}O$ reconstructions, which decreases with increasing sampling resolution.
Precision errors of all $\Delta_{47}$-based approaches eventually converge with the initially much lower precision
error of $\delta^{18}O$ reconstructions when sampling resolution increases. However, the sampling resolution that is
required for $\Delta_{47}$-based reconstructions to rival or outcompete the $\delta^{18}O$ reconstructions differs, with
**optimization** requiring lower sampling resolutions than the other methods (e.g. 20-40 samples/year
compared to 40-80 samples year for **smoothing** and **binning**; **Fig. 8A-D**). Accuracy also decreases with
sampling resolution (**Fig. 8E-H**). When grouping all cases together, it becomes clear that $\delta^{18}O$
reconstructions can only approach the accuracy of $\Delta_{47}$-based approaches for reconstructions of MAT.
Seasonality in both SST and $\delta^{18}O_{sw}$ is most accurately reconstructed using **binning**, and the **smoothing**
approach once again performs worst.

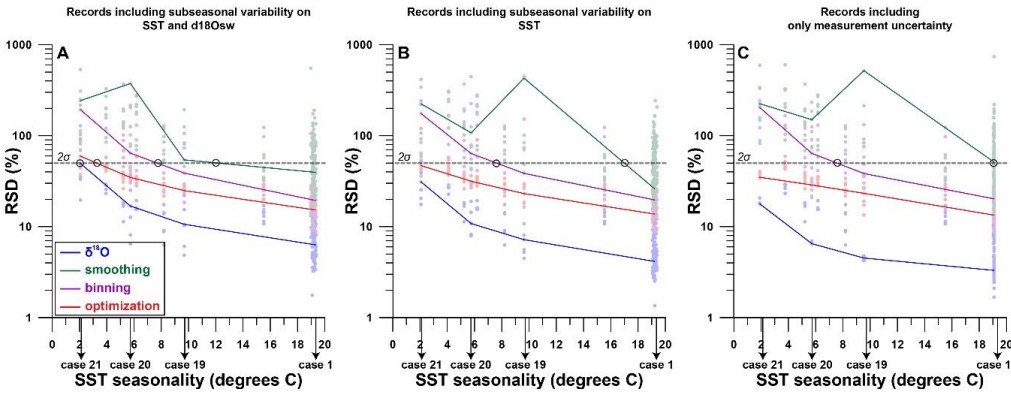

***Figure 9***: *Effect of SST seasonality range (difference between warmest and coldest month) in the record*
*on the relative precision of SST seasonality reconstructions (one standard deviation divided by the mean*
*value). Panel **A** shows precision results if random variability ("weather patterns") in both SST and $\delta^{18}O_{sw}$*
*as well as measurement uncertainty is added to the records (see **3.1.1** and **S1**). Panel **B** shows precision*
*of records with random variability in SST and measurement uncertainty only. Panel **C** shows precision if*
*only measurement uncertainty is considered. Color coding follows the scheme in **Fig. 1** and **Fig. 4**. Shaded*
*dots represent results at various sampling resolutions, while bold lines are averages for all reconstruction*
*approaches. Black circles highlight the places where curves cross the threshold of two standard deviations,*
*which indicates the minimum SST seasonality that can be resolved within 2 standard deviations (~95%*
*confidence level) using the reconstruction approach.*





**4.4 Resolving SST seasonality**
Comparison of cases 19, 20 and 21 (SST seasonality of 9.7°C, 5.7°C and 2.1°C respectively) with control
case 1 (SST seasonality of 19.3°C) allowed us to study how changes in the seasonal SST range affect the
precision of measurements (**Fig. 9**; see also **S1**). The data reconfirms that $\delta^{18}O$ reconstructions are most
precise; a deceptive statistic given the risk of highly inaccurate results this approach yields (see **Fig. 7**).
Taking into consideration only analytical uncertainty, all approaches except for **smoothing** can confidently
resolve at least the highest SST seasonality within a significance level of two standard deviations (~95%)
using a moderate sampling resolution (mean of all resolutions shown in **Fig. 10**). Increasing sampling
resolution improves the precision of $\Delta_{47}$-based reconstructions (see **Fig. 8D**), so high sampling resolutions
(0.1 or 0.2 mm) allow smaller seasonal differences to be resolved. When random sub-annual variability is
added to the SST and $\delta^{18}O_{sw}$ records (see 3**.1.3** and **S1**), the minimum seasonal SST extent that can be
resolved decreases for all approaches (**Fig. 9B** and **9C**). Nevertheless, $\delta^{18}O$ and **optimization**
reconstructions remain able to resolve a relatively small SST seasonality of 2-4°C, even with all noise added
to the records.



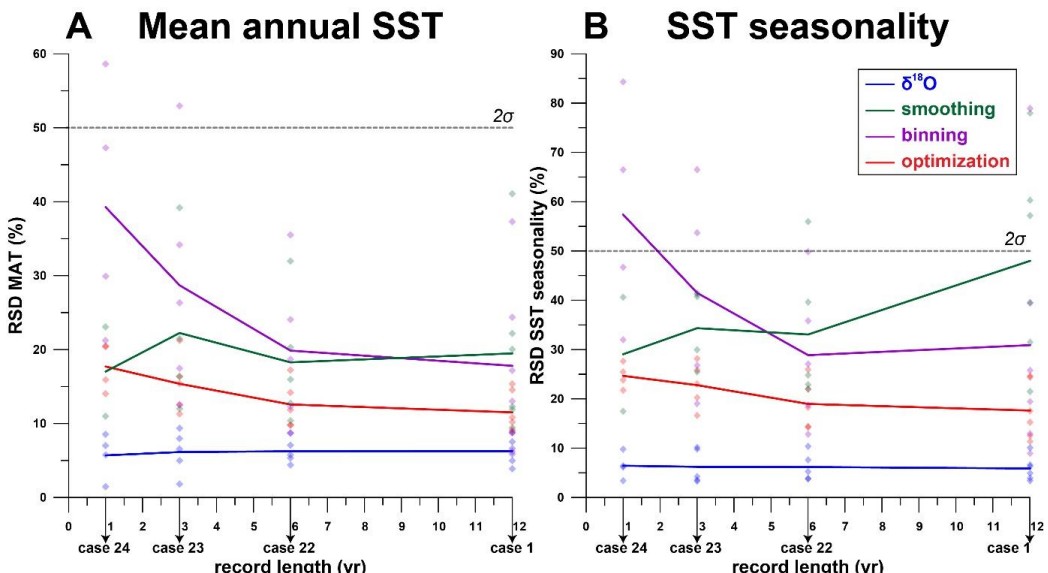

***Figure 10**: Effect of record length (in years) on the relative precision (one standard deviation as fraction of*
*the mean value) of results of reconstructions of mean annual SST (**A**) and SST seasonality (**B**). Shaded*
*dots represent results for the six different sampling resolutions. Solid lines connect averages for cases 1,*
*22, 23 and 24 for each reconstruction approach. Color coding follows the scheme in **Fig. 1** and **Fig. 4**.*
**4.5 Effect of record length**
The effect of variation in the length of the record was investigated by comparing cases 22, 23 and 24 (record
length of 6 years, 3 years and 1 year) with the control case (record length of 12 years; see **Fig. 10** and **S1**).
As expected, the precision of MAT and SST seasonality results slightly increases in larger datasets (longer
records). However, this pattern is not clear in **smoothing** and **δ18O** reconstructions. The difference between
reconstruction approaches remains relatively constant regardless of the length of the record, with general
precision hierarchy remaining intact (**δ18O** > **optimization** > **binning** > **smoothing**). An exception occurs
in the case of very short records (1-2 years), where the **smoothing** gains an advantage over other $\Delta_{47}$-
based methods due to its lack of sensitivity to changes in the record length. For very short (<3 yr) records,
**binning** reconstructions are not precise enough to resolve MAT and SST seasonality within two standard
deviations (~95% confidence level). Most of the variation in precision with record length is driven by very
high precision errors of reconstructions based on records with low sampling resolutions (sampling intervals





of 1.55 mm or 3.25 mm; see also **Fig. 8A-D**). As a result, most of the reduction in precision in shorter
records can be mitigated by denser sampling.

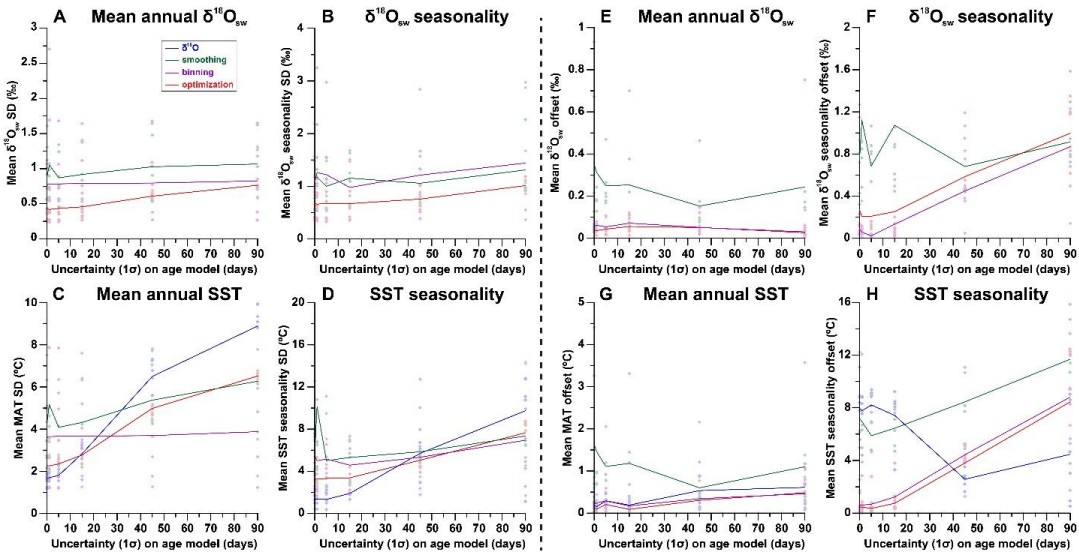


*Figure 11*: *Effect of uncertainty in age model on the reproducibility (standard deviation on estimate) of*
*results of reconstructions of mean annual $\delta^{18}O_{sw}$ (**A**) and seasonal range in $\delta^{18}O_{sw}$ (**B**), mean annual SST*
*(**C**) and seasonal range in SST (**D**). Effect of uncertainty in age model on the accuracy (offset from true*
*value) of results of reconstructions of mean annual $\delta^{18}O_{sw}$ (**E**) and seasonal range in $\delta^{18}O_{sw}$ (**F**), mean*
*annual SST (**G**) and seasonal range in SST (**H**). Color coding follows the scheme in **Fig. 1** and **Fig. 4**.*

### 4.6 Effect of age model uncertainty

Uncertainty on the age model has a significant effect on both the precision and the accuracy (**Fig. 11**) of
reconstructions using all approaches. The **$\delta^{18}O$** reconstructions are most strongly affected by uncertainties
in the age model and suffer from a large decrease in precision with increasing age model uncertainty (**Fig.**
**11C-D**). The high reproducibility of the **$\delta^{18}O$** approach in comparison with the $\Delta_{47}$ approaches quickly
disappears when age model uncertainty increases beyond 20-30 days. Interestingly, the accuracy of SST
seasonality reconstructions based on **$\delta^{18}O$** initially improves with age model uncertainty (**Fig. 11H**).
However, this observation is likely caused by the fact that age model uncertainty was compared based on
conditions in case 9, which features a phase offset between SST and $\delta^{18}O_{sw}$ seasonality causing the **$\delta^{18}O$**
method to be highly inaccurate even without age model uncertainty. The precision of **smoothing** and





**optimization** approaches also decreases with increasing age model uncertainty (**Fig 11A-D**), and the
**optimization** approach loses its precision advantage over the **binning** and **smoothing** approaches when
age model uncertainty increases beyond 30 days. The monthly **binning** approach is very robust, and its
precision does not significantly decrease with increasing age model uncertainty. Seasonality
reconstructions through both the **binning** and **optimization** approach quickly lose accuracy when age
model uncertainty increases. The accuracy of the **smoothing** approach remains the worst of all approaches
in regardless of age model uncertainty (**Fig. 11E-H**).






**Figure 12**: *Overview of averages and ranges of accuracy (absolute offset from real value) and precision (one standard deviation from the mean) on mean annual $\delta^{18}O_{sw}$ (**A**) and seasonal range in $\delta^{18}O_{sw}$ (**B**), mean annual SST (**C**) and seasonal range in SST (**D**) within all cases using the four different reconstruction approaches. Color coding follows the scheme in **Fig. 1** and **Fig. 4**. Box-whisker plots for precision and accuracy cross at their median values and outliers (colored symbols) are identified based on 2x the interquartile difference (see **Fig. 6 and 7**)*



## 5. Discussion

### 5.1 Performance of reconstruction approaches

#### *5.1.1 $\delta^{18}O_c$ vs $\Delta_{47}$-based reconstructions*

A summary of the general reliability of the four approaches is shown in **Figure 12**. The **$\delta^{18}O$** reconstructions are generally less accurate than $\Delta_{47}$-based reconstructions (especially **binning** and **optimization**; see **Fig 12** and **S10**). This is a consequence of the assumption that $\delta^{18}O_{sw}$ remains constant year-round, and that we know its true value. Both these assumptions are problematic in absence of independent evidence of the value of $\delta^{18}O_{sw}$, especially in deep time settings (see e.g. Veizer and Prokoph, 2015; Henkes et al., 2018). The risk of this assumption is made clear when comparing cases in which $\delta^{18}O_{sw}$ is indeed constant year-round at the assumed value (0‰VSMOW; e.g. cases 1-6 and 19-24) with cases in which shifts in $\delta^{18}O_{sw}$ occur, especially when these shifts are out of phase with respect to the SST seasonality (e.g. cases 9-11, 18 and 25-33; **Fig. 7C-D**). Cases mimicking or based on natural SST and SSS variability (cases 14-18 and 30-33) as well as the modern oyster data (**Fig. 5**) yield stronger inaccuracies in MAT and seasonality reconstructions, showing that even in many modern natural circumstances the assumption of constant $\delta^{18}O_{sw}$ is problematic.

It is important to consider that the value of mean annual $\delta^{18}O_{sw}$ remained very close to the assumed value of 0‰VSMOW (within 0.15‰) in all cases except for natural data cases 30 (-1.55‰VSMOW), 32 (1.01‰VSMOW; see **S5**) and the real oyster data (-1.42‰VSMOW; **Fig. 5**). The SST values of these cases reconstructed using $\delta^{18}O_c$ data show large offsets from their actual values (+6.7°C, -4.7°C and +10.3°C for case 30, case 32 and the real oyster data respectively; see **Fig. 5 and 7C** and **S5**). These offsets are equivalent the temperature offset one might expect from inaccurately estimating $\delta^{18}O_{sw}$ (~-4.6 °C/‰VSMOW; Kim and O'Neil, 1997) and are only rivaled by the offset in reconstructions of case 18 (+5.0°C), which has growth hiatuses obscuring the coldest half of the seasonal cycle. The fact that such differences in $\delta^{18}O_{sw}$ exist even in modern environments should not come as a surprise, given the available data on variability of $\delta^{18}O_{sw}$ (at least -3‰ to +2‰VSMOW; e.g. LeGrande and Schmidt, 2006) and SSS (30 to 40 ; ESA, 2020) in modern ocean basins. However, it should warrant caution in using $\delta^{18}O_c$ data for SST reconstructions in modern settings. Implications for deep time reconstructions are even greater, given the



uncertainty on and variability in global average (let alone local) $\delta^{18}O_{sw}$ values (Jaffrés et al., 2007; Veizer
and Prokoph, 2015). The complications of using $\delta^{18}O_c$ as a proxy for marine temperatures in deep time are
discussed in detail in O'Brien et al. (2017), andTagliavento et al. (2019).
The analytical uncertainty of individual $\delta^{18}O_c$ aliquots (typically 1 S.D. of 0.05‰; e.g. de Winter et al., 2018)
represents only ~1.1% of the variability in $\delta^{18}O_c$ over the seasonal cycle (~4.3‰ for the default 20°C
seasonality in case 1, following Kim and O'Neil, 1997). This is much smaller than the analytical uncertainty
of $\Delta_{47}$ (typically 1 S.D. of 0.02-0.04‰; e.g. Fernandez et al., 2018; de Winter et al., 2020), which equates
to 25-50% of the seasonal variability in $\Delta_{47}$ (~0.08‰ for 20°C seasonality, following Bernasconi et al., 2018;
see **S8**). This roughly 20-fold difference in relative precision causes $\delta^{18}O_c$ based SST reconstructions to be
much more precise (see **Figs 6**, **8-11**) than those based on $\Delta_{47}$, and forces the necessity for grouping $\Delta_{47}$
data in reconstructions. However, as discussed above, the low precision of **$\delta^{18}O$** reconstructions is
misleading and not a useful statistic if they are highly inaccurate.
Our results show that paleoseasonality reconstructions based on $\delta^{18}O_c$ can only be relied upon if there is
strong independent evidence of the value of $\delta^{18}O_{sw}$ and if significant sub-annual variability in $\delta^{18}O_{sw}$ (>0.3‰,
equivalent to a 2-3°C SST variability; see **Fig. 8-9**; Kim and O'Neil, 1997) can be neglected with confidence.
Examples of such cases include fully marine environments unaffected by influxes of (isotopically light)
freshwater or evaporation (increasing $\delta^{18}O_{sw}$; Rohling, 2013). Carbonate records from suitable
environments include, for example, the *A. islandica* bivalves from considerable depth (30-50m) in the open
marine Northern Atlantic (e.g. Schöne et al., 2005, on which case 33 is based). Previous reconstruction
studies show that $\delta^{18}O_{sw}$ in smaller basins such as the Western Interior Seaway are heavily influenced by
the processes affecting $\delta^{18}O_{sw}$ on smaller scales (e.g. Petersen et al., 2016). Consequently, accurate
quantitative reconstructions of seasonal range in shallow marine environments with extreme seasonality
may not be feasible using the **$\delta^{18}O$** approach, because these environments are invariably characterized by
significant fluctuations in $\delta^{18}O_{sw}$ and growth rate.
While variability in $\delta^{18}O_{sw}$ compromises accurate $\delta^{18}O$-based seasonality reconstructions, the compilation
in **Fig. 3** shows that its influence on the $\delta^{18}O$ records is too small to affect the shape of the record to such
a degree that seasonality is fully obscured. While natural situations with $\delta^{18}O_{sw}$ fluctuations large enough



to totally counterbalance the effect of temperature seasonality on δ18O records are imaginable, these cases
are likely rare. This means that chronologies based on δ18O seasonality, which are a useful tool to anchor
seasonal variability in absence of independent growth markers (e.g. Judd et al., 2018), are reliable in most
natural cases.
*5.1.2 Seasonality reconstructions using moving averages (**smoothing**)*
Of the three methods for combining $\Delta_{47}$ data, the **smoothing** approach clearly performs worst in all four
reconstructed parameters (MAT, SST seasonality, mean annual δ18O$_{sw}$ and δ18O$_{sw}$ seasonality), both in
terms of accuracy and precision (**Fig. 12**). While applying a moving average may be a good strategy for
lowering the uncertainty of $\Delta_{47}$-based temperature reconstructions in a long time series (e.g. Rodríguez-
Sanz et al., 2017), the method underperforms in cases where baseline and amplitude of a periodic
component, spike or event (e.g. MAT and SST seasonality) are extracted from a record. This is likely due
to the smoothing effect of the moving average, which reduces the seasonal cycle and causes highly
inaccurate seasonality reconstructions (offsets mounting to >6°C; **Fig. 12**). This bias is especially
detrimental in cases where the seasonal cycle is obscured by seasonal growth halts (e.g. case 18), multi-
annual trends in growth (e.g. case 4, 14 and 17) and multi-annual trends in SST (e.g. case 15 and 17; see
**Fig. 6** and **Fig. 7**). The lack of performance of the **smoothing** approach can be slightly mitigated by
increasing sampling resolution (**Fig 8**), but even at high sampling resolutions (every 0.1 or 0.2 mm) the
method still fails to reliably resolve seasonal SST ranges below 5°C even in idealized cases (case 19-21;
**Fig. 9**). Increasing the number of samples by analyzing longer records does not improve the result, because
smoothing of the seasonal cycle by a moving average window introduces the same dampening bias as long
as the temporal sampling resolution (number of samples per year) remains equal (**Fig. 10**).
More critically, employing the **smoothing** method may give the illusion that seasonality is more reduced,
and severely bias reconstructions. This bias highlights the importance of using the official meteorological
definition of seasonality as the difference between the averages of warmest and coldest month in
paleoseasonality work (O'Donnell et al., 2012). This definition is much more robust than the "annual range"
often cited based on maxima and minima in δ18O$_c$ records. This "annual range" strongly depends on
sampling resolution, which is typically <12 yr$^{-1}$ (Goodwin et al., 2003), equivalent to the third lowest sampling





interval (0.75 mm) simulated in this study. Therefore, we strongly recommend future studies to adhere to
the monthly definition of seasonality to foster comparison between studies. While inter-annual variability is
lost by combining data from multiple years into estimates of WMMT and CMMT, this approach increases
precision, accuracy and comparability of paleoseasonality results. Inter-annual variability can still be
discussed from plots of raw data against age or depth.
*5.1.3 Monthly **binning**, sample size **optimization** and age model uncertainty*
Overall, the most reliable paleoseasonality reconstructions can be obtained from either **binning** or
**optimization** (**Fig. 12**). In general, **optimization** is slightly more precise, while **binning** yields more
accurate estimates of seasonal range in SST and $\delta^{18}O_{sw}$ (**Fig. 12B and D**). The more flexible combination
of aliquots in the **optimization** routine yields improved precision (especially on mean annual averages) in
cases where parts of the record are undersampled or affected by hiatuses and simultaneous fluctuations
in both SST and $\delta^{18}O_{sw}$ (e.g. case 3-6, 14---18, 30-33). The downside of this flexibility is that in case of
larger sample sizes, the seasonal variability may be dampened, like in the **smoothing** approach (see **5.1.1**).
The rigid grouping of data in monthly bins in **binning** prevents this dampening and therefore yields slightly
more accurate estimates of seasonal ranges in SST and $\delta^{18}O_{sw}$. A caveat of **binning** is that it requires a
very reliable age model of the record, as least on a monthly scale. If the age model has a large uncertainty,
there is a risk that samples are grouped in the wrong month, which compromises the accuracy of **binning**
reconstructions, especially for reconstructions of seasonal range (**Fig 11H**).
Techniques for establishing independent age models for climate archives range from counting of growth
layers or increments (Schöne et al., 2008; Huyghe et al., 2019), modelling and extracting of rhythmic
variability in climate proxies through statistical approaches (e.g. De Ridder et al., 2007; Goodwin et al.,
2009; Judd et al., 2018) and interpolation of uncertainty on absolute dates (e.g. Scholz and Hoffman, 2011;
Meyers, 2019; Sinnesael et al., 2019). While propagating uncertainty in the data on which age models are
based onto the age model is relatively straightforward, errors on underlying *a priori* assumptions such as
linear growth rate between dated intervals, (quasi-)sinusoidal forcing of climate cycles and the uncertainty
on human-generated data such as layer counting are very difficult to quantify (e.g. Comboul et al., 2014).
The uncertainty of such age models of climate records is thus difficult to assess and may not be normally





distributed. A simplified test of the effect of a normally distributed error on the age value of each proxy data
point (case 25-29) shows that uncertainties in the age domain can significantly compromise reconstructions
(**Fig. 11**). Within the scope of this study, only the effect of symmetrical, normally distributed uncertainties
on an artificial case with phase decoupled SST and $\delta^{18}O_{sw}$ seasonality (case 9) was tested. The effect of
other types of uncertainties on other cases remains unknown, highlighting an unknown uncertainty in
paleoseasonality and other high-resolution paleoclimate studies that may introduce bias or lead to over-
optimistic errors on reconstructions. Future research could aim to quantify this unknown uncertainty by
propagating estimates of various types of uncertainty on depth values of samples and on the conversion of
depth to time in age models.
**5.2 Conditions influencing success of reconstructions**
Our results show that the reliability (accuracy and precision) of SST and $\delta^{18}O_{sw}$ reconstructions depends
on case-specific conditions. The range of cases tested in this study allowed us to evaluate the effect of
variability in SST, growth rate, $\delta^{18}O_{sw}$, sampling resolution and record length relative to the control case
(case 1; see **S1**). A summary of the effects of these changes is given in **Table 1**.





| Variable | cases | Metric | Effect on reconstructions | | | |
|---|---|---|---|---|---|---|
| | | | $\delta^{18}O$ | smoothing | binning | optimization |
| SST | 12 15 17 19-21 30-33 | Precision | 0 | +++ | + | 0 |
| | | Accuracy | + | + | 0 | + |
| Growth rate | 2-6 14-18 30-33 | Precision | + | ++ | ++ | + |
| | | Accuracy | + | ++ | 0 | + |
| $\delta^{18}O_{sw}$ | 7-11 13-18 30-33 | Precision | + | ++ | 0 | 0 |
| | | Accuracy | +++ | +++ | + | ++ |
| Sampling resolution | 1-33 | Precision | 0 | +++ | ++ | ++ |
| | | Accuracy | + | + | +++ | + |
| Record length | 22-24 | Precision | 0 | 0 | +++ | ++ |
| | | Accuracy | + | 0 | ++ | ++ |
| Age model uncertainty? | 25-29 | Precision | +++ | ++ | 0 | ++ |
| | | Accuracy | + | + | ++ | ++ |

***Table 1****: Qualitative summary of the effects of changes in variables relative from the ideal case on*
*reconstructions using the four approaches. The "cases" column lists cases in which the changes in the*
*respective variable relative to the control case (case 1) were represented (see **S1**). "0" = negligible effect,*
*"+" = weak increase in uncertainty, "++" = moderate increase in uncertainty, "+++" = strong increase in*
*uncertainty. Details on the precision and accuracy of all tests is given in **S12**.*

*5.2.1 SST variability*
Variability in water temperature most directly affects the proxies under study. By default (case 1), SST is
taken to vary sinusoidally around a MAT of 20°C with an amplitude of 10°C (see 3.**1.1**, **Fig. 2** and **S1**). In
case of exceptions, in which multi-annual variability in SST is simulated (e.g. case 15 and 17), the accuracy
of SST reconstructions using **$\delta^{18}O$** and **optimization** are reduced, while the **binning** approach is less
strongly affected. Examples of such multi-annual cyclicity are El-Niño Southern Oscillation (ENSO;
Philander, 1983) or North Atlantic Oscillation (NOA; Hurrell, 1995). The effect is especially large in case 17,
which simulates a tropical environment with reduced SST seasonality and a strong multi-annual cyclicity.
This type of environment is analogous to the environment of tropical shallow water corals, which are often
used as archives for ENSO variability (e.g. Charles et al., 1997; Fairbanks et al., 1997). As such, these
virtual records should be analogous to tropical cases from the Australian Great Barrier Reef (case 31) and
Red Sea (case 32; see **Fig. 6-7**). We therefore recommend future researchers to use the **binning** approach



on carbonate records where multi-annual cyclicity is prevalent and if a reliable age model can be
established for these records (as in e.g. Sato, 1999; Scourse et al., 2006; Miyaji et al., 2010).
*5.2.2 Growth rate variability and hiatuses*
**Figures 6 and 7** show that variations in the growth rate of records, including the occurrence of hiatuses,
have a strong effect on reconstructions, especially using the **smoothing** approach. In general, hiatuses
and slower growth reduce precision of monthly SST and $\delta^{18}O_{sw}$ reconstructions by reducing mean temporal
sampling resolution (samples/yr; see **Fig. 8**), and because specific parts of the record are undersampled.
The effect on accuracy depends strongly on the timing of changes in growth rate or the occurrence of
hiatuses. Cases 26 simulate specific growth rate effects and can be used to test these differences. The
**smoothing** method is especially sensitive to changes in growth rate that take place in specific seasons,
such as hiatuses in winter (case 2) or summer (case 3) and growth peaks in summer (case 5) or spring
(case 6). The other reconstruction approaches are less affected by this bias, because they generally do not
mix samples from different seasons and therefore produce less smoothing. The **$\delta^{18}O$** method is especially
well suited to deal with changes in growth rate because it does not require combining different aliquots for
accurate SST reconstructions. The **binning** and **optimization** approaches are slightly less accurate in
cases where growth rate decreases linearly or seasonally along the entire record (cases 46; **Fig. 5**). This
likely occurs because these two methods consider all samples in the records at once, instead of only a
subset at any one time (as in the **smoothing** method), and are therefore more sensitive to changes in
temporal sampling resolution along the record. It is worth noting that **optimization** is especially sensitive to
sharp changes in growth rate in summer (e.g. cases 11, 14, 16 and 17) because those conditions force the
**optimization** routine to use larger sample sizes or include samples outside the warmest month for summer
temperature estimates.
A worst-case scenario of reconstructions hampered by growth rate variability and hiatuses is represented
by case 18, where the entire cold half of the year is not recorded. Such cases result in strong biases in
reconstructions of mean annual and seasonal ranges in SST and $\delta^{18}O_{sw}$, regardless of which method is
used. In such extreme cases the record simply contains insufficient information to reconstruct variability in
growth rate, SST and $\delta^{18}O_{sw}$, and it seems that no statistical method would enable this missing information





to be recovered. In such cases, the only way to eliminate bias in reconstructions would be to establish
reliable age models, independent of $\delta^{18}O$ or $\Delta_{47}$ data, which show that a large part of the seasonal cycle is
missing.
While hiatuses encompassing half of the seasonal cycle are uncommon, changes in growth rate are
common in accretionary carbonate archives because conditions for (biotic or abiotic) carbonate
mineralization often vary over time. This variability is either driven by biological constraints, such as
senescence (e.g. Schöne, 2008; Hendriks et al., 2012), reproductive cycle (Gaspar et al., 1999) or stress
(Surge et al., 2001; Compton et al., 2007) or by variations in the environment that promote or inhibit
carbonate production, such as seasonal variations temperature (Crossland, 1984; Bahr et al., 2017) or
precipitation (Dayem et al., 2010; Van Rampelbergh et al., 2014). In general, such conditions occur more
frequently in mid- to high-latitude environments than in low-latitudes, and in more coastal environments
rather than in open marine settings, because these environments contain stronger variations in the factors
that influence growth rates (e.g. temperature, precipitation or freshwater influx; e.g. Surge et al., 2001;
Ullmann et al., 2010). This difference was simulated the cases representing natural variability (case 14-18
and 30-33), with accuracy in the coastal high-latitude settings (cases 16, 18 and 29) more strongly affected
by changes in growth rate. Because in such highly variable environments growth rate variability often co-
occurs with variability in $\delta^{18}O_{sw}$, using $\delta^{18}O_c$-based reconstructions is not advised, unless $\delta^{18}O_{sw}$ variability
can be constrained or neglected (which is rare in these environments). An additional complication is that
growth rate variability cannot always be resolved because of uncertainties in the record's age model (see
**4.1.3**). Therefore, reconstructions in these highly dynamic environments may not allow all variables that
introduce bias to be isolated, and irregular variability in growth rate and $\delta^{18}O_{sw}$ will invariably introduce
uncertainty in SST reconstructions, even when applying the best $\Delta_{47}$-based approaches (e.g. **binning** and
**optimization**). In such examples, the results of natural variability cases (14-18 and 30-33) and of the real
oyster data (**Fig. 5**) may serve as benchmarks for the degree of uncertainty that may remain unexplained
in these records.



*5.2.3 Variability in $\delta^{18}O_{sw}$*
Large increases in uncertainty on reconstructions are caused by variations in $\delta^{18}O_{sw}$ (see **Fig. 6 and 7**). As
discussed in **4.1.1**, these variations have a large effect on the accuracy of $\delta^{18}O_c$-based reconstructions,
and their occurrence constitutes the main advantage of applying the $\Delta_{47}$ thermometer (Eiler, 2011).
However, results of cases 7-11 in **Fig. 7** and **Table 1** show that $\delta^{18}O_{sw}$ variations can also bias $\Delta_{47}$-based
reconstructions, especially those of seasonal ranges and using the **smoothing** approach. **Smoothing**
reconstructions are biased by these $\delta^{18}O_{sw}$ shifts in much the same way as they are affected by shifts in
growth rate (see **4.1.2**). The **optimization** approach, especially when used for reconstructions of $\delta^{18}O_{sw}$
seasonality, is sensitive to seasonal changes in $\delta^{18}O_{sw}$ in antiphase with SST seasonality and by increases
in $\delta^{18}O_{sw}$ in summer (e.g. due to excess evaporation). This effect arises because the **optimization**
approach orders data based on $\delta^{18}O_c$ and $\Delta_{47}$ seasonality to isolate the $\delta^{18}O_{sw}$-SST relationship. Both
antiphase $\delta^{18}O_{sw}$ seasonality and summer evaporation dampens the seasonal $\delta^{18}O_c$ cycle and therefore
influences the reconstruction of the $\delta^{18}O_{sw}$-SST relationship. A good example of this is seen in the real
oyster data (**Fig. 5**), where $\delta^{18}O_{sw}$ and SST vary in phase and $\delta^{18}O_{sw}$ dampens the SST seasonality. The
**binning** approach is more robust against $\delta^{18}O_{sw}$ variability that dampens the seasonal cycle and is therefore
a better choice for absolute SST reconstructions in environments where summer evaporation or other
$\delta^{18}O_{sw}$ variability in phase with SST seasonality is expected to occur, if the age model is reliable enough to
allow monthly binning of raw data (see **4.1.3**). Indeed, reconstructions from the lagoonal environment (case
16) and Red Sea case (case 32 which is characterized by strong summer evaporation; e.g. Titschack et
al., 2010) show that **binning** is the most reliable choice in these environments.
*5.2.4 Variability in sampling resolution and record length*
Other factors influencing the effectivity of reconstructions are the sampling resolution and the length of the
record. Many of the cases discussed in this study represent idealized cases with comparatively high
sampling resolutions over comparatively long (12 yr) paleoseasonality records, which yield large sample
sizes. By comparison, the typical age of mollusks, which are often used as paleoseasonality archives, is 2-
5 years (Ivany, 2012). Records with the highest sampling resolutions (0.1 and 0.2 mm) contain up to 1200
samples. This is not an unfeasible number of samples, but it is highly unlikely to be applied in paleoclimate



studies given the limitation of resources (e.g. instrument time) and the desire to analyze multiple records
from different specimens, species, localities or ages to gain a better understanding of the variability in
paleoseasonality (e.g. Goodwin et al., 2003; Schöne et al., 2006; Petersen et al., 2016). In some cases
large datasets are meticulously collected from single carbonate records (e.g. Schöne et al., 2005;
Vansteenberge et al., 2016; de Winter et al., 2020a; Shao et al., 2020). However, in such studies, the aim
is often to investigate variability at a higher (e.g. daily; de Winter et al., 2020a) resolution or longer
timescales (e.g. decadal to millennial; Schöne et al., 2005; Vansteenberge et al., 2016; Shao et al., 2020)
in addition to the seasonal cycle, rather than to improve the reliability of reconstructing one type of variability
(e.g. seasonality) alone. In this study, extreme (sometimes unnatural, e.g. case 18) cases were chosen
deliberately to explore the effect of different conditions and guide researchers in deciding their sampling
strategy to optimize their samples and resources in function to their various research goals.
**Fig. 8** shows that increasing temporal sampling resolution (samples/yr) improves both the accuracy and
precision of all $\Delta_{47}$-based reconstructions. This occurs because $\Delta_{47}$ samples have a large analytical
uncertainty (see **5.1.2**) and grouping of data therefore improves reconstructions. Interestingly, in $\delta^{18}O_c$-
based reconstructions precision decreases with increasing sample size while accuracy increases (**Fig. 8C-**
**D**). This is explained by the fact that the analytical uncertainty of $\delta^{18}O_c$ measurements is much smaller than
the variability introduced by natural sub-annual variability in SST and $\delta^{18}O_{sw}$ unrelated to the seasonal cycle
(see **S4**). Therefore, higher sampling resolutions allow $\delta^{18}O_c$ records to better capture this sub-seasonal
variability, which introduces more noise on the seasonal cycle (reducing precision) but causes monthly
mean SST and $\delta^{18}O_{sw}$ to be more accurately reconstructed. Towards higher sampling resolutions, the gap
in precision between $\delta^{18}O_c$- and $\Delta_{47}$-based reconstructions closes, eventually (in an ideal case) diminishing
the advantage of high analytical precision in $\delta^{18}O_c$ measurements (**Fig. 8C-D**).
The rate of increase in precision and accuracy with sampling resolution is not the same for each method,
and an optimum sample resolution can be defined for each method after which improving sampling
resolution does not significantly improve the reliability of the reconstruction (as in de Winter et al., 2017).
**Figure 8** shows that this optimum is different depending on which variable (MAT, SST seasonality, mean
annual $\delta^{18}O_{sw}$ or $\delta^{18}O_{sw}$ seasonality) is reconstructed. Therefore, **Fig. 8** will allow future researchers to





determine the sampling resolution that is tailored to their purpose. In general, the improvement after a
sample size of 20-30 samples per year is negligible for the **binning** and **optimization** methods if the total
number of samples (depending on both sampling resolution and record length) is sufficient for monthly
temperature reconstructions. Our data show that 200-250 paired $\delta^{18}O_c$ and $\Delta_{47}$ measurements are in
general sufficient for a standard deviation of 2-3°C on monthly SST reconstructions using the **binning** or
**optimization** approach (**Fig. 8**; **S5**).
Record length only has a minimal influence on the **optimization** method but for very short records (≤2
years) **binning** becomes very imprecise, especially at low sampling resolutions (**Fig. 10**). The reason for
this is that the sample size within monthly time bins becomes too small in these cases, while the more
flexible sample size window of the optimization routine circumvents this problem. The choice between these
two approaches should therefore be based on a tradeoff between the length of the record (in time) and the
number of samples that can be retrieved from it. As a result, shorter-lived, fast-growing climate archives,
such as large or fast-growing (e.g. juvenile) mollusk shells, are best sampled using a high temporal
resolution (30+ samples/yr) sampling strategy with the **optimization** approach. Longer lived archives with
a lower mineralization rate, such as annually laminated speleothems, corals and gerontic mollusks, are
best sampled using long time series at monthly resolution using the **binning** approach.
A simplified decision tree that could guide sampling strategies for future paleoseasonality studies is shown
in **Figure 13**. Note that choices and tradeoffs for these reconstructions may differ depending on the archive
and environment in which it formed (see discussion above).

## Schematic guide to reconstructing SST and $\delta^{18}O_{sw}$ from accretionary carbonate archives

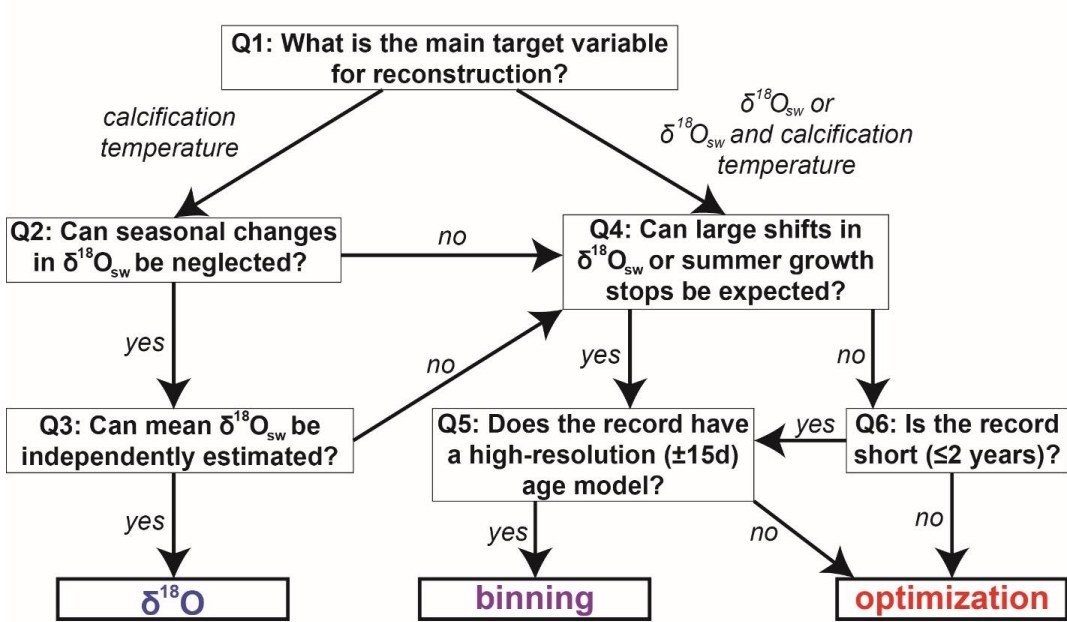


**Figure 15**: *Schematic guide to choosing the right approach for reconstructing annual mean or seasonality in SST and $\delta^{18}O_{sw}$ from accretionary carbonate archives. Recommendations are based on the results of testing all four approaches on the entire range of cases. Researchers can follow the six steps (questions Q1-6) to decide on the right approach for reconstructing the target variable. Guidelines are based on minimizing both accuracy and precision (see details in **S11**). Note that the **smoothing** approach is never the best choice. The choice between the two remaining $\Delta_{47}$-based approaches (**binning** and **optimization**) relies heavily on the situation and may be driven by a preference for more accurate or more precise results.*

### 5.3 Implications for clumped isotope sample size

The **optimization** technique for grouping $\Delta_{47}$ aliquots for accurate SST and $\delta^{18}O_{sw}$ reconstructions allows

us to assess the limitations of the clumped isotope thermometer for temperature reconstructions from high-

resolution carbonate archives. The actual optimal sample size given by the approach is different for different

cases and depends on the temporal sampling resolution and the characteristics of the record (see **S5**). As

expected, in cases more similar to the ideal case (case 1), optimal sample sizes are low (~14--24), while

sample sizes quickly increase in more complicated cases based on simulated natural environments (case

14--18) or cases based on actual SST and SSS data (cases 30-33). More confined SST seasonality (cases

19-21) also requires larger samples to reconstruct (up to 100 samples in some cases). This is not surprising,



because variability within samples will increase in more complicated records in which the seasonality is
smaller or more obscured by other environmental variability. The optimal sample size between cases and
sampling resolutions is not normally distributed but tails towards high sample sizes with some extreme
outliers (Shapiro Wilk test p << 0.05; **S12**). The median sample size of all our simulations is 17 aliquots.
This number lies between the minimum number of 14 ~100 µg replicates of standards calculated by
Fernandez et al. (2017) and the minimum of 20-40 ~100 µg aliquots required for optimal paleoseasonality
reconstruction from fossil bivalves by de Winter et al. (2020a). This is to be expected since many of the
cases explored in this study represent ideal cases compared with the natural situation. However, in many
cases a measure of random sub-annual variability in SST and $\delta^{18}O_{sw}$ was added (see **S2**), simulating a
more realistic environment and resulting in poorer precision than replicates of a carbonate standard (as in
Fernandez et al., 2017). Our simulations show that the optimum number of samples to be combined in
seasonality studies depends on both the analytical uncertainty of $\Delta_{47}$ measurements (as represented by
the estimate in Fernandez et al., 2017) and the variability between aliquots pooled within a sample that is
attributed to actual variability within the record (as represented by our simulations and the estimate in de
Winter et al. 2020a). The optimal sample size is therefore a good measure for the limitations of temperature
variability that can be resolved in a record. As such, this number, together with the overview in **S1**, can help
researchers decide which strategy to apply for combining measurements to obtain the most reliable
paleoseasonality estimates, or to decide whether extra sampling is required, even if the chosen approach
is not to use the **optimization** routine itself.
**5.4 Implications for other sample size problems**
While the discussion above focuses on optimizing approaches for combining samples for clumped isotope
analyses in paleoseasonality reconstructions, the problem of combining samples to lower uncertainty and
isolate variation in datasets is very common (e.g. Zhang et al., 2004; Merz and Thieken, 2005; Tsukakoshi,
2011). Therefore, the approaches outlined and tested in this study have applications beyond
paleoseasonality reconstructions. Below, we briefly highlight four examples of problems that could benefit
from applying similar approaches for lowering the uncertainty of estimates of target variables or reducing
the number of analyses required to meet analytical requirements.





*5.4.1 Tooth bioapatite*
Enamel from vertebrate teeth constitute a useful archive for paleoenvironmental and paleoecological
change in the terrestrial realm, complementing the carbonate records discussed in this work (e.g. Luz and
Kolodny, 1985; Fricke et al., 1996; Balasse, 2002; Van Dam and Reichart, 2009; de Winter et al., 2016).
However, the tooth bioapatite archive suffers from similar limitations of sample size and resolution as
carbonate archives when it comes to reconstructing high-resolution variability (see discussion in Passey
and Cerling, 2002 and Kohn, 2004). Oxygen and carbon isotopes of carbonate and phosphate in tooth
enamel contain valuable information about the animal's life cycle and environment (e.g. Fricke et al., 1996;
Balasse et al., 2002; Van Dam and Reichart, 2009). However, structurally bound carbonate constitutes a
mere 2-5% of tooth enamel (LeGeros et al., 1986), and enamel samples need to be pretreated to remove
labile components, so analyses of $\delta^{18}O$ in these archives require comparatively large sample sizes (0.5-1
mg; Fricke et al., 1998; Balasse, 2002; Pellegrini and Snoeck, 2016). Phosphate-bound $\delta^{18}O$ is less
susceptible to diagenesis, but requires a more complicated procedure to analyze, resulting in similar sample
size limitations (Joachimski et al., 2002; Lecuyer et al., 2007). Most applications of isotope profiles from
teeth rely on precise determination of both the phase and amplitude of the seasonal cycle, and therefore
suffer from the same complications as isotope records in carbonate archives (e.g. Balasse et al., 2002;
Straight et al., 2004). The **binning** and **optimization** approaches discussed here could help reduce
uncertainty and provide a basis for better comparison of seasonal profiles in tooth enamel.
*5.4.2 Cyclostratigraphy*
Within the field of cyclostratigraphy, a multitude of stratigraphical approaches have been developed for
signal processing, with the aim to use regular orbital cycles expressed in stratigraphic time series as tools
for dating rock sequences (e.g. Paillard et al., 1996; Meyers, 2014; Sinnesael et al., 2016). However, the
focus on timing has caused many methods for extracting the climatic impact of these orbital cycles from
stratigraphic records (e.g. bandpass filtering; Hilgen, 1991) to remain qualitative. This is unfortunate,
because the magnitude of the effect of this cyclicity on climate and environmental change is of major interest
in paleoclimatology studies (e.g. Berger, 1992; Shackleton, 2000; Zachos et al., 2001; Lourens et al., 2005;
De Vleeschouwer et al., 2017a). The problem of quantitatively extracting the impact of orbital cycles is very



similar to the problem of paleoseasonality reconstructions central to this study, and the same approaches
can therefore be used in the orbital time domain. The time **binning** approach is probably most robust for
this purpose, since cyclostratigraphic records are often longer (record length >> period of the cycle) and
sampling resolutions (samples/cycle) are often lower than in seasonal records (see **5.2.4**; e.g. De
Vleeschouwer et al., 2017b). Quantitative analyses of the contribution of orbital cyclicity to rhythmic
changes in paleoclimate can help separate variability in records caused by external forcing from autocyclic
behavior or (positive or negative) feedback of the climate system itself (Lourens et al., 2010; Noorbergen
et al., 2017; Nohl et al., 2018).
*5.4.3 Strontium isotope dating*
Another type of analysis that could benefit from smart combination of measurement results is strontium
isotope dating. The strontium isotope composition ($^{87}Sr/^{86}Sr$) of the ocean has evolved over time, and the
isotopic composition of marine carbonates can therefore be used to estimate the age of the sample by
comparing it with a composite strontium isotope curve (Elderfield, 1986; McArthur et al., 2012). In time
intervals where the global marine strontium isotope curve is steep, strontium isotope dating ranks among
the most precise methods for absolute dating in stratigraphy (Wegreich et al., 2012). However, accurate
dating based on the strontium isotope curve requires propagation of errors on the composite curve and the
sample. Doing so results in asymmetric errors due to the non-linear character of the strontium isotope
curve, which require complex error propagation (see Barlow, 2003; 2004; Wan et al., 2019). The state-of-
the-art uncertainty of individual strontium isotope analyses ranges between 210 ppm (1 standard deviation;
Yobregat et al., 2017), which translates to an age uncertainty of 100-200 kyr, (1 standard deviation)
depending strongly on the slope of the global strontium isotope curve at the time interval under study.
Combining multiple strontium isotope analyses from the same stratigraphic unit can reduce the uncertainty
on these composite ages (Korte and Ullmann, 2016; de Winter et al., 2020b), allowing the dating method
to be combined with cyclostratigraphy to produce for orbital scale age models (see **5.4.2**). In stratigraphy
studies that use this dating method, the need arises to compromise between the resolution of the age model
and the precision and accuracy of dating, analogous to the tradeoff that occurs when combining $\Delta_{47}$
analyses for paleoseasonality reconstructions outlined in this study. In this case, the **smoothing** approach





with a dynamic moving window discussed in this study is likely the best candidate for combining data to
improve these age models. Such an approach can be seen as a more flexible adaptation of the $\Delta_{47}$-based
approach for SST reconstruction outlined in Rodríguez-Sanz et al. (2017) that provides the flexibility to
adapt the sample window depending on the available data and the slope of the global strontium curve. At
the same time, the shape of the global composite strontium isotope curve itself can be refined by using a
similar protocol on well-dated samples. The approaches discussed in this study are more adaptable to
changes in sampling density over time and can in theory achieve higher precision than the LOWESS fit
approach currently employed for constructing the global composite (McArthur et al., 2012). Similarly,
techniques for compromising between sampling resolution and accuracy and precision can be applied to
improve other dating methods based on matching curves such as radiocarbon dating (Ramsay and Lee,
2013), carbon isotope stratigraphy (Salzman and Thomas, 2012) and dendrochronology (Cook and
Kairiukstis, 2013).
*5.4.4 Sub-seasonal variability*
Ultra-high-resolution records from fast-growing archives (e.g. mollusks) are an emerging phenomenon in
the field of high-resolution paleoclimatology (e.g. Sano et al., 2012; Warter and Müller, 2017, de Winter et
al., 2020a). The emergence of such records allows new information to be obtained about the daily cycle
(Warter et al., 2018; de Winter et al., 2020a) and extreme weather events (Yan et al., 2020) in the past,
potentially bridging the gap between weather and climate reconstructions. The sampling resolution required
to resolve variability at such a fine temporal scale warrants an even more careful consideration of the
tradeoff between sample size, sampling resolution and analytical uncertainty than the paleoseasonality
examples considered here. If quantitative estimates of insolation, temperature and the frequency of extreme
weather events are to be reconstructed from these novel records, a compromise will need to be found
between analytical uncertainty and the temporal resolution of measurements (Sano et al., 2012; de Winter
et al., 2020a; Yan et al., 2020). Applying the temporal (e.g. hourly) binning method (**binning**) discussed
here on long, (sub-)daily resolved records could yield more accurate and precise records of ultra-high-
resolution variability, given its reliability in extracting accurate cycle amplitude (e.g. seasonality) from long,
less densely sampled records (see **5.1.3**). Fast-growing bivalve and gastropod shells have already been





marked as promising archives for such variability, while other fast-growing archives such as *Acropora* corals
remain to be explored (Bak et al., 2009; Strauss et al., 2014; de Winter et al., 2020c). It must be noted that
models for the timing of carbonate deposition in accretionary carbonate archives at the sub-daily scale are
highly uncertain and that this may complicate the use of the **binning** approach (see **5.1.3**), in which case
**optimization** may be more appropriate.
*5.4.5 Event stratigraphy*
Accurate and precise temperature reconstructions of short-lived (10-100 kyr) episodes of climate change
present a problem comparable to resolving seasonality in paleoclimate archives. Examples of such events
include the Mesozoic ocean anoxic events (Hesselbo et al., 2000; Jenkyns, 2010), early Paleogene
hyperthermals (Stap et al., 2010; Lauretano et al., 2015, 2018) and stepwise climate perturbations such as
the Eocene-Oligocene transition (Dupont-Nivet et al., 2007; Lear et al., 2008) studied in deep-sea records.
Currently, reconstructions of temperature variability in the deep-sea during such events are based on
benthic foraminiferal $\delta^{18}O_c$ (e.g. Erbacher et al., 2001; Lui et al., 2009; Stap et al., 2010; Lauretano et al.,
2015, 2018), but may not be reliable due to assumptions made on $\delta^{18}O_{sw}$. Deep-sea sedimentary
environments are generally characterized by low sedimentation rates (~1 cm/kyr) as well as low abundance
and small size of microfossils (e.g. foraminifera) which serve as archives of past marine conditions (e.g.
Stap et al., 2010; Jennions et al., 2015). This limits the number of aliquots that can be obtained for $\Delta_{47}$ and
other analyses through these climate events. In these studies, a **smoothing** approach would probably
underestimate the 'true' amplitude of temperature or geochemical change. With sufficient record length and
perhaps by combining multiple events, **binning** or **optimization** based on proxy data would be the most
accurate and precise approach to resolve transient temperature change in the deep-sea during the
geological past.



## 6. Conclusions and recommendations

The reliability of three $\Delta_{47}$-based approaches to reconstruct seasonality from accretionary carbonate archives was evaluated in comparison with the conventional $\delta^{18}O_c$-based reconstructions in a wide range of case studies. From the results, we conclude that while $\delta^{18}O_c$-based reconstructions (**$\delta^{18}O$**) yield superior precision for SST reconstructions, this method runs a high risk of yielding inaccurate results due to innate assumptions about the value of $\delta^{18}O_{sw}$, which has to be estimated and assumed constant year-round. Unless a $\delta^{18}O_{sw}$ can be independently constrained or variability in $\delta^{18}O_{sw}$ can be neglected, $\Delta_{47}$-based reconstructions should be the method of choice for absolute mean annual temperature and SST seasonality reconstructions. Various techniques for combining $\Delta_{47}$ data were evaluated. Our findings suggest that smoothing $\Delta_{47}$ data using a moving average (**smoothing**) results in almost all cases in a dampening of the seasonal cycle which severely hampers recovery of seasonality. Applying the **smoothing** approach results in inaccuracies in reconstructions of MAT as well, especially in cases where part of the seasonal cycle is obscured by variability in growth rate or multi-annual trends. More reliable seasonality reconstructions are achieved with two approaches for combining $\Delta_{47}$ data using time binning (**binning**) or applying a flexible sample size optimization (**optimization**) approach. Of these two approaches, **optimization** achieves better precision and can resolve smaller seasonal temperature differences with confidence. However, **binning** is often more accurate, and outperforms **optimization** as the most reliable approach. This is especially true in cases with growth stops or $\delta^{18}O_{sw}$ changes in phase with temperature seasonality (e.g. strong seasonal evaporation or freshwater influx) and in longer multi-annual time series with a reliable age model. **Optimization** is the better choice for shorter (<3 years) records, especially if the sampling resolution can be increased, such as in short, fast growing climate archives.

Despite the distinct focus on the problem of resolving seasonality in carbonate archives, the findings in this study have applications for other problems where sample size and sampling resolution put limits on the ability to resolve specific trends, events and cycles from time series. Examples include, but are not limited to, resolving sub-annual variability in geochemical records from tooth bioapatite, quantifying the impact of orbital cycles on paleoclimate, refining strontium isotope dating by strategic sample combination, resolving daily scale variability and weather patterns in ultra-high-resolution climate records and quantifying the





impact of climate events in the geological record. While the above mentioned recommendations of the
**optimization** and **binning** methods are likely valid for most studies aiming to quantify the mean and
amplitude of a specific cycle or event (equivalent to MAT and SST seasonality), (dynamic) moving averages
(**smoothing**) are expected to yield the best results in studies quantifying aperiodic trends from longer data
series.

**Code availability**
Annotated R scripts used to make calculations for this study are available in the digital supplement uploaded
to the open-source online repository Zenodo (www.doi.org/10.5281/zenodo.3899926).

**Data availability**
Supplementary data, figures and tables as well as all scripts used to do the calculations and create the
virtual datasets used in this study are deposited in the open-source online repository Zenodo
(www.doi.org/10.5281/zenodo.3899926).

**Author contributions**
NJW designed the study, wrote the scripts for all calculations, and created a first draft of the manuscript
text and figures. MZ, TA and NJW worked together from the first draft towards the final manuscript. All
authors contributed to the representation of the data and methods in figures and to the discussion of the
implications of the data in the discussion.

**Competing Interests**
The authors have no potential conflicts of interest to declare with regards to this study.



## Acknowledgements

The authors would like to thank all members of the Clumped Isotope research group of Utrecht University, most notably Ilja Kocken and dr. Inigo Müller, for their comments and recommendations on a presentation of the initial results of this study.

## Financial support

NJW is funded by the European Commission through a Marie Sklodowska Curie Individual Fellowship (UNBIAS, grant # 843011) and by the Flemish Research Council (FWO) through a Junior Postdoctoral Fellowship (12ZB220N).

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
