# Peer review of "Optimizing sampling strategies in high-resolution paleoclimate records"

_Climate of the Past, 2020_

## Referee Comment (RC1) · Anonymous Referee #1 · 24 Nov 2020

In 'Optimizing sampling strategies in high-resolution paleoclimate records', de Winter et al. propose four data treatment methods for constraining (paleo)climatological parameters from sequentially-sampled isotopic records and interrogate the accuracy and precision of each of these approaches using a combination of real and virtual datasets. Seasonally-resolved proxy data (e.g.,  $\delta$ 180 and more recently  $\Delta$ 47) offer the potential to provide greater clarity into past climates. However, a number of factors complicate the interpretation of these archives. Conventional oxygen isotope thermometry requires an assumption regarding the isotopic composition of seawater, which, as the authors point out, is both spatially and seasonally variable even in the modern ocean. Clumped isotope thermometry is independent of seawater composition, circumventing this limitation, but comes with a much larger analytical uncertainty (made even larger

by the small sample sizes required for sub-annual resolution) and thus the data can be noisy. By exploring how these and several other factors, such as sampling size and length of record, impact the accuracy and precision of recovered climatological information, the authors are able to develop a framework for determining the best sampling and statistical strategies to maximize the fidelity and minimize the uncertainty of these recovered parameters.

This study serves to significantly advance sclerochronologic research by providing critical insights into the uncertainties surrounding seasonal analyses and a quantitative, statistically-rooted means of extracting useful climate information from often noisy records. Overall, the paper is well-written, the experimental design well thought out, and the discussion thorough. The topic is pertinent and will be of interest to a broad paleoclimatological audience. I recommend only minor revisions, most of which pertain to improving the clarity and readability of the manuscript.

General comments:

(1) Based on Figs. 6, 7, and 12 it seems like the  $\delta$ 18O reconstructions aren't necessarily less accurate than the other approaches, but that the accuracy of the results are more variable. The approach accurately reconstructs MAT in Cases 1-14 and 19-29 and accurately reconstructs seasonal range in Cases 1-6, 19-24, and 30-33. Significant deviations occur only when the mean annual  $\delta$ 18Osw differs significantly from the assumed value and/or when there is a strong seasonality to  $\delta$ 18Osw (which, as you point out, is a realistic and often ignored scenario).

Given these findings, I'm curious why you have not attempted to combine the  $\delta$ 180 approach with another method to maximize both precision and accuracy of the results. If, for example, you were to use the binning or optimization of clumped isotope technique to constrain the seasonal seawater cycle, you could then apply those results to the higher-precision  $\delta$ 180 approach and alleviate the assumption of normal-marine/invariant seawater composition. For a real-world example of this approach see

CPD
Keating-Bitonti et al. (2011), who use bulk summer and winter clumped samples to identify a summertime freshwater influx impacting their oxygen isotope values.

(2) Though beyond the scope of the variables considered here, there are two additional complicating factors in real-world sclerochronologic data that may be worth mentioning at some point: (2a) Unequal sample time-averaging as a function of growth rate in practical application (versus virtual subsampling), the number of days (or weeks or months) averaged into a single sample will vary; when the organism is growing quickly the sample will represent comparatively less time than when it is growing slowly, using the same diameter drill bit. If growth slowdowns/hiatuses, e.g., correspond with winter extremes, this results in not only fewer but also more time averaged (and thus dampened) estimates of winter temperature. (2b) Uncertainty in the seasonal phasing of SSTs in paleoclimate studies - even in instances were samples from a fossil specimen can be aligned along some reliable age model (be it via growth band counting or a statistical model), assigning those data points a calendar date is ambiguous at best. The timing of the date of maximum and minimum SST can vary considerably based on latitude, environment, and other local factors. This is a particularly important source of uncertainty to consider when binning the data by month – shifting the calendar date assumptions by, e.g., 15 days, would result in a whole new grouping of monthly data points and could significantly alter the results.

(3) Regarding the figures, I really appreciate the consistent color theme throughout, but for accessibility purposes you may want to double check that the color scheme is colorblind safe. Additionally, there is a lot of really useful information that is not always easily extracted from the figures – I'd encourage you to think about creative ways of graphically presenting the information that will allow the reader to quickly glean the most important aspects. For example, you could include a heatmap showing how accuracy and precision vary by case and climatological parameters (similar to the colored conditional formatting of S12), which may be easier to interpret than the line plots.

(4) I'm not convinced that Section 5.4 (Implications for other sample size problems)
sufficiently contributes to the overall content of the manuscript to warrant inclusion. This section reads like a grant proposal rather than a discussion and could easily be condensed into a single one-paragraph section briefly outlining potential additional applications of the approach.

(5) Prior to the final submission, the manuscript, supplement, and figures all require a thorough read through to ensure consistency of terminology, case numbers, and color scheme. For example, high precision/accuracy are at times conflated with high reproducibility error/offset in the text. Case numbers are not always consistent (both in the text and supplements), and it appears as if the colors have been switched in at least one figure. I've tried to point out examples of these in the line-by-line comments, but I'm sure that I've missed some. The supplemental files are hard to navigate - it would be helpful to include a 'Read Me' tab in applicable excel files defining all acronyms and abbreviations and descriptions of the information provided in each subsequent tab.

Specific Comments.

\*NOTE: line numbers restart after L347; second set of line numbers indicated by LN

Text:

L12: The term 'events' is a bit ambiguous here, particularly since the manuscript focuses on recovering climatological parameters (i.e., multi-year averages that smooth 'events')

L54: 'the seasonal cycle is the most important cycle in Earth's climate.' This is a bold and rather subjective statement - I can think of many who might argue the carbon cycle is equally important. I'd suggest changing 'most' to 'one of the most' or omitting the statement all together.

L91: Colon missing after 'Optimization'

L120: Consider changing 'depth domain' to 'sampling domain' (here and in all subsequent references); ontogenetic trajectories aren't necessarily depths

CPD
L121: Delete 'this'

L139: There's an error in this equation ( $\delta$ 18Osw,freshwater is repeated twice). The mass balance should read:  $\delta$ 18Osw = f\*  $\delta$ 18Osw,freshwater + (1-f)\* $\delta$ 18Osw,ocean

L142: Space between the per mille symbol and VSMOW; this information is repeated on L145

L157: Change 'is' to 'are'

L159, etc.: I'd suggest stating upfront that all references to seawater composition are in reference to VSMOW; the VSMOW reference after each composition is a bit clunky

L217: Should be case 31 not 30, I believe

L272: Was the seasonal seawater composition range calculated based on the difference between warmest and coldest month (as described in the text) or between the most depleted and enriched  $\delta$ 18Osw seawater months? There are many examples where the SST and seawater cycles are out of phase (e.g., springtime snow melt flux driving a  $\delta$ 18Oseawater depletion extreme prior to the summer temperature extreme). The difference between peak summer and winter seawater composition is an important variable to constrain, but it does not necessarily always equate to the seasonal range in  $\delta$ 18Osw.

L285: Is it fair to assume a normal marine in this environment? I suspect that this was done to illustrate a point and while I don't disagree that constraining mean annual (let alone seasonal variability) in  $\delta$ 180sw in deep time adds huge uncertainty to conventional oxygen isotope interpretations, we can often make somewhat more realistic estimates of the (mean annual) value based on latitude and environment than just the global normal marine value.

L294: Lower accuracy or higher offset

L294: Change 'on' to 'of'
LN14: Accuracy improves, offset decrease

LN14: Change 'samples year' to 'samples/year'

LN128: Add a space between 'and' and 'Tagliavento'

LN136: Higher precision or lower reproducible error

LN163: I'm unclear how 'event or spike' relate to the examples discussed here; I suggest omitting for clarity.

LN194: Change 'as' to 'at'

LN231: Modify 'case of exceptions, in which' to 'cases in which' for clarity

LN248: Change 'Cases 26' to 'Cases 2-6'

LN380,382: Are the double hyphens (between 14-24 and 14-18) intentional?

LN456: 'between 210 ppm' is not a range, I think a hyphen or a second number is missing

Figures:

Figure 2: Consider adding heading labels for the blocks of virtual cases (e.g., a 'Sensitivity cases' header for 1-13, 'Natural cases' for 14-18, etc.). It will help focus the reader's eye to the differences between cases and will serve as a more useful reference throughout the results and discussion.

Figure 7: Color scheme is off in this figure – it looks like the smoothing and binning data may be reversed in the box plots.

Figure 9: Please define RSD in the caption (the definition is there, but it is never directly linked to the acronym)

Supplement:

S12: Case 18 is missing (Natural case 5); numbering of all subsequent cases is off
References cited:

Keating-Bitonti, C.R., Ivany, L.C., Affek, H.P., Douglas, P. and Samson, S.D., 2011. Warm, not super-hot, temperatures in the early Eocene subtropics. Geology, 39(8), pp.771-774.

---

## Referee Comment (RC2) · Anonymous Referee #2 · 31 Jan 2021

In this manuscript De Winter et al present a complex study evaluating sampling and statistical methodologies aimed at constraining uncertainties in the application of stable isotope data to reconstruct past sea water temperature. Given the rapid rate at which new sclerochronological stable isotope records are being constructed, such an investigation is both timely and pertinent. Whilst this manuscript could provide an extremely useful outline of methodologies for future isotope studies, it is extremely difficult to follow. This is not helped by the nomenclature used and jumping between virtual and real isotope data and different methodologies without sufficient explanation for what these methodologies are and why they are being used. The readability of both text and figures therefore needs to be improved. The main area of weakness in this manuscript is associated with the description of the methodology. The authors need to set out in far

clearer terms the exact application of data in each of the three analytical processes and exactly which data is used. Given the readability and lack of clarity in the methodology, I would recommend major revisions be made to this manuscript prior to publication Other comments: Why was mean annual temperature used as a target? When isotope records are developed from species that have a known growth hiatus, for example during winter, it is unlikely that mean annual temperature would be targeted. An alternative target would be used, for example mean summer or a growing season mean instead. Targeting annual mean would make the record look worse than it actually is. Pg 5 ln 89-91: The explanations for each of the methods are insufficient. Line 195: "not exactly normally distributed" remove the word exactly. Pg 29 ln 119: missing "to" between "equivalent the" Pg 30 ln 142: 144 – "Carbonate records from suitable environments include, for example, the A. islandica bivalves from considerable depth (30-50m) in the open marine Northern Atlantic (e.g. Schöne et al., 2005, on which case 33 is based)." It would b inaccurate to assume that there is no variability in ïA̧ďʹ18Osw in NE Iceland. Variability in ïA̧ďʹ18Osw can also derive from changes in water mass that are bathing the shells during carbonate precipitation.

Pg 30 ln 146: "the processes affecting $\delta$18Osw on smaller scales" such as? Pg 30: 151-152: "While variability in $\delta$18Osw compromises accurate $\delta$18O-based seasonality reconstructions, the compilation in Fig. 3 shows that its influence on the $\delta$18O records is too small to affect the shape of the record to such a degree that seasonality is fully obscured."

Pg 36 ln 270 "While hiatuses encompassing half of the seasonal cycle are uncommon" I would not say that they are uncommon. There are many examples of sub-tropical to polar marine bivalve species which exhibit a cessation in growth during certain months of the year.

Pg 36 ln 280 missing "in" between "simulated the"

Sections 5.4.1 to 5.4.5 should be deleted. This is already a very long manuscript and

this extra detail does not add anything to the focus of this manuscript.

All figures: Colours used in Figures need to be changed to be accessible. The website https://colorbrewer2.org/ provides a useful free resource to check colour choices. Fig 2: Whilst it is good to see a visual representation of each of the cases, the size of the panels makes it difficult to actually see what the purpose of each case is. It would be helped to add a schematic or table highlighting the purpose of each case. Fig 4A. Whilst a schematic would be a helpful figure to help the reader visualise what is a very complex methodology, the schematic presented in Fig 4A doesn't help. Currently this schematic does not help to clarify the methodology. I would suggest that the example provided in Fig 4V is removed and more space provided for panel A. This would provide space to add detail along with the arrows. Fig 6. Axis text size needs to be increased. Fig 12: Change the colours on the plot, you can't tell the difference between the d18O and the binning results.

---

## Editor Comment (EC1) · Alberto Reyes (Editor) · 31 Jan 2021

Dear Dr. de Winter and co-authors, Your manuscript has now been reviewed by two referees, and the public commentary period is closed. By now you have likely received an automatic email requesting that you provide a response to the referee comments.

Note that you do not submit a revised manuscript when uploading your responses. However, you may find it useful to frame these responses around the revisions you anticipate making to the manuscript. Both referees indicated the work will likely be a useful addition to the sclerochronology literature, so I do anticipate inviting you to submit a revised manuscript after reviewing your responses to referee comments.

Sincerely, Alberto Reyes (handling editor)

---

## Author Comment (AC1) · 3 Feb 2021

Dear Alberto Reyes, dear reviewers,

We would like to thank both reviewers for their constructive comments on our manuscript. We are confident that their suggestions will help us improve our manuscript and will try our best to implement them during our revision. The reviewers made a range of minor suggestions, which we will address in a point-by-point rebuttal on resubmission. Below, we briefly summarize our strategy for revising the manuscript in response to the more substantial points raised by the reviewers.

Methodology – The major concern by Reviewer #2 seems to be our presentation of the methodology for our reconstruction approaches. We will revise the Aim and Methods sections of the manuscript such that the different reconstruction approaches are better explained on first mention and will include more detail in these explanations to clarify the difference between the approaches. In addition, we have now compiled all R scripts used for creating virtual data and reconstructions into a documented R package which is available through the online open-source R database CRAN (https://cran.r-project.org/web/packages/seasonalclumped). Following Reviewer #2, we will adapt Figure 4, splitting it up into two figures to allow for more room for the flow chart illustrating how the virtual datasets are created. If this split causes the number of figures to be too high, we are happy to move the flow chart with the example of Case 31 to the supplement, although we prefer to keep it in the main text.

Definitions – Both reviewers raise valid concerns about our use of definitions and references for cases, reliability benchmarks (e.g. accuracy and reproducibility) and methodology. To improve the clarity of our explanations, we will go through the manuscript in detail to make sure all our terminology and references to cases and reconstruction approaches are consistent and well-defined.

Benchmarks for testing – Both reviewers raised questions about our use of benchmarks for mean annual and seasonality against which to test our reconstructions. We would like to clarify that we use the range in monthly $\delta$18Osw (most enriched minus most depleted month) as our benchmark for seasonality in $\delta$18Osw. We retain that our use of mean annual temperature as benchmark is justified. Even though many studies reconstruct "mean growing temperature", the ultimate goal of climate reconstruction is to obtain more information about climate variables independent from the archive. In our view the conversion to mean annual temperature must be made eventually and this presents a source of uncertainty which we wanted to include in our analysis. Finally, our use of 0‰ VSMOW as a benchmark for mean annual $\delta$18Osw does not overestimate inaccuracies of the $\delta$18Oc-method, since almost all our virtual datasets are based on variability around this value. If anything, this $\delta$18Osw assumption underestimates the real inaccuracy of $\delta$18Oc reconstructions because in many cases the mean annual

$\delta$18Osw value is not known in the fossil domain and its estimates may be much farther from the true value.

Data presentation – Both reviewers made suggestions on how to improve the way in which we present the data generated in our study. In reply to the suggestion by reviewer #1, we will modify Figures 6 and 7 (overviews of accuracy and reproducibility of all cases) to make it more intuitive to spot the differences between cases at first glance. We like the idea of using a color-coded heatmap to visualize this complex data and will experiment with this concept. The original Figures 6 and 7 will be retained in the supplement. In response to concerns from both reviewers, we will adapt the colour scheme used throughout the manuscript to make it more accessible to colour-blind readers. We appreciate the suggestion by Reviewer #2 of using the Colorbrewer tool and we will use this tool to select our colours.

Outlook to future research – Both reviewers agree that our somewhat lengthy outlook chapter (section 5.4) makes the (already quite complex) manuscript too long. We therefore opt to follow the suggestion of Reviewer #1 to summarize the entire section 5.4 into one paragraph to save space.

---

## Author Comment (AC2) · 5 Feb 2021

Dear Alberto Reyes, dear reviewers,

In addition to the more general Author Reply we submitted earlier in this discussion, please find our detailed point-by-point rebuttal to the comments raised by the reviewers attached. We hope that our replies to the reviewers' concerns and the revisions we propose will be adequate to merit publication of our manuscript in Climate of the Past.

Kind regards,

Niels J. de Winter

on behalf of all authors

[Figure]

Please also note the supplement to this comment:
https://cp.copernicus.org/preprints/cp-2020-118/cp-2020-118-AC2-supplement.pdf

———————————————————

[Figure]

**Supplement:**

Dear Alberto Reyes, dear reviewers,

In addition to the more general Author Reply we submitted earlier in this discussion, please find a detailed point-by-point rebuttal to the comments raised by the reviewers below. We hope that our replies to the reviewers' concerns and the revisions we propose will be adequate to merit publication of our manuscript in Climate of the Past. In the text below, we indicate review comments in red and our replies in **black**.

**Anonymous Referee #1**

*In 'Optimizing sampling strategies in high-resolution paleoclimate records', de Winter et al. propose four data treatment methods for constraining (paleo)climatological parameters from sequentially-sampled isotopic records and interrogate the accuracy and precision of each of these approaches using a combination of real and virtual datasets. Seasonally-resolved proxy data (e.g., δ18O and more recently Δ47) offer the potential to provide greater clarity into past climates. However, a number of factors complicate the interpretation of these archives. Conventional oxygen isotope thermometry requires an assumption regarding the isotopic composition of seawater, which, as the authors point out, is both spatially and seasonally variable even in the modern ocean. Clumped isotope thermometry is independent of seawater composition, circumventing this limitation, but comes with a much larger analytical uncertainty (made even larger by the small sample sizes required for sub-annual resolution) and thus the data can be noisy. By exploring how these and several other factors, such as sampling size and length of record, impact the accuracy and precision of recovered climatological information, the authors are able to develop a framework for determining the best sampling and statistical strategies to maximize the fidelity and minimize the uncertainty of these recovered parameters.*

*This study serves to significantly advance sclerochronologic research by providing critical insights into the uncertainties surrounding seasonal analyses and a quantitative, statistically-rooted means of extracting useful climate information from often noisy records. Overall, the paper is well-written, the experimental design well thought out, and the discussion thorough. The topic is pertinent and will be of interest to a broad paleoclimatological audience. I recommend only minor revisions, most of which pertain to improving the clarity and readability of the manuscript.*

*General comments:*

*(1) Based on Figs. 6, 7, and 12 it seems like the δ18O reconstructions aren't necessarily less accurate than the other approaches, but that the accuracy of the results are more variable. The approach accurately reconstructs MAT in Cases 1-14 and 19-29 and accurately reconstructs seasonal range in Cases 1-6, 19-24, and 30-33. Significant deviations occur only when the mean annual δ18Osw differs significantly from the assumed value and/or when there is a strong seasonality to δ18Osw (which, as you point out, is a realistic and often ignored scenario).*

*Given these findings, I'm curious why you have not attempted to combine the δ18O approach with another method to maximize both precision and accuracy of the results. If, for example, you were to use the binning or optimization of clumped isotope technique to constrain the seasonal seawater cycle, you could then apply those results to the higher-precision δ18O approach and alleviate the assumption of normalmarine/invariant seawater composition. For a real-world example of this approach see Keating-Bitonti et al. (2011), who use bulk summer and winter clumped samples to identify a summertime freshwater influx impacting their oxygen isotope values.*

This is a valid suggestion by the reviewer, and we agree that a combination of the low-precision clumped isotope method with (potentially) lower accuracy $\delta^{18}O_c$ measurements could yield an optimal compromise between the pros and cons of these methods. In fact, our "optimization" approach is based on this idea. However, the approach of combining bulk clumped isotope measurements and high-resolution $\delta^{18}O_c$ measurements has disadvantages, especially in cases where the $\delta^{18}O_{sw}$

seasonality is in antiphase with temperature seasonality (see comment on line 272 below). In addition, larger bulk measurements for clumped isotope analyses may significantly average temperature and $\delta^{18}O_{sw}$ reconstructions of summer and winter seasons compared to microsampled transects. We will clarify the advantages and disadvantages of different methods for combining clumped and $\delta^{18}O_c$ measurements in seasonality studies by introducing the approach by Keating-Bitonti et al. (2011) in the introduction and comparing it with our approaches in the discussion (sections 5.1).

*(2) Though beyond the scope of the variables considered here, there are two additional complicating factors in real-world sclerochronologic data that may be worth mentioning at some point: (2a) Unequal sample time-averaging as a function of growth rate - in practical application (versus virtual subsampling), the number of days (or weeks or months) averaged into a single sample will vary; when the organism is growing quickly the sample will represent comparatively less time than when it is growing slowly, using the same diameter drill bit. If growth slowdowns/hiatuses, e.g., correspond with winter extremes, this results in not only fewer but also more time averaged (and thus dampened) estimates of winter temperature. (2b) Uncertainty in the seasonal phasing of SSTs in paleoclimate studies – even in instances were samples from a fossil specimen can be aligned along some reliable age model (be it via growth band counting or a statistical model), assigning those data points a calendar date is ambiguous at best. The timing of the date of maximum and minimum SST can vary considerably based on latitude, environment, and other local factors. This is a particularly important source of uncertainty to consider when binning the data by month – shifting the calendar date assumptions by, e.g., 15 days, would result in a whole new grouping of monthly data points and could significantly alter the results.*

These are very good suggestions and we gladly incorporate these considerations in our revised discussion. Since both these complications relate to the timing of growth of sampled intervals, we will incorporate them into the section discussing the effects of uncertainties in the age model (section 5.2.2).

*(3) Regarding the figures, I really appreciate the consistent color theme throughout, but for accessibility purposes you may want to double check that the color scheme is colorblind safe. Additionally, there is a lot of really useful information that is not always easily extracted from the figures – I'd encourage you to think about creative ways of graphically presenting the information that will allow the reader to quickly glean the most important aspects. For example, you could include a heatmap showing how accuracy and precision vary by case and climatological parameters (similar to the colored conditional formatting of S12), which may be easier to interpret than the line plots.*

We appreciate the suggestion by the reviewer and paid special attention to the colour scheme of our figures, which was also a concern raised by Referee #2. We now use the RdYlBu 4-colour scheme provided by Colorbrewer 2.0 to make our figures colour-blind- and print-friendly (see https://colorbrewer2.org/?type=diverging&scheme=RdYlBu&n=4). To be honest, the multi-dimensional character of our results caused us some difficulties in coming up with a proper presentation method for aggregating all results in overview figures (Figures 6 and 7). We appreciate the reviewer's suggestion for revising our presentation method for these figures and agree that a colormap may be a more intuitive way to visualize this data. We will experiment with this idea and try to come up with a more reader-friendly visualization of the data now shown in figures 6 and 7.

*(4) I'm not convinced that Section 5.4 (Implications for other sample size problems) C3 sufficiently contributes to the overall content of the manuscript to warrant inclusion. This section reads like a grant proposal rather than a discussion and could easily be condensed into a single one-paragraph section briefly outlining potential additional applications of the approach.*

Both reviewers expressed doubts about the added value of section 5.4 in its current form. We therefore decided to shorten this section into one paragraph, as suggested by Referee #1 as opposed to deleting the entire section following Referee #2.

*(5) Prior to the final submission, the manuscript, supplement, and figures all require a thorough read through to ensure consistency of terminology, case numbers, and color scheme. For example, high precision/accuracy are at times conflated with high reproducibility error/offset in the text. Case numbers are not always consistent (both in the text and supplements), and it appears as if the colors have been switched in at least one figure. I've tried to point out examples of these in the line-by-line comments, but I'm sure that I've missed some. The supplemental files are hard to navigate - it would be helpful to include a 'Read Me' tab in applicable excel files defining all acronyms and abbreviations and descriptions of the information provided in each subsequent tab.*

This is a very valid concern, and we apologize for any mistakes that may have crept in our terminology or colour scheme in the initial version of the manuscript. For the revision, we will go through the manuscript, figures and supplements in detail and make sure all references and colours are consistent. In response to this comment and related comments by Referee #2, we will also more clearly define our use of terminology for describing reliability of reconstructions ("accuracy", "precision", "offset", etc.) and provide more detail in the supplements explaining the data tables. In addition, we have now compiled the R scripts of our reconstruction approaches into an R package. This package, including documentation of all scripts, is now freely available online through the CRAN database (https://cran.r-project.org/web/packages/seasonalclumped) to ensure easy access to the reader. We will cite the package in the revised version of the manuscript.

*Specific Comments.*

*\*NOTE: line numbers restart after L347; second set of line numbers indicated by LN*

Our apologies for the break in the line numbering, we will amend this in the revised version.

*Text:*

*L12: The term 'events' is a bit ambiguous here, particularly since the manuscript focuses on recovering climatological parameters (i.e., multi-year averages that smooth 'events')*

We acknowledge that this statement may be ambiguous because the length of an "event" may depend on its (broad) definition. To keep referring to the compromise between achieving accurate climate information by combining measurements on the one hand, and retaining a high temporal resolution in reconstructions on the other hand, we propose to rephrase as follows:

"The challenge is to isolate meaningful information on climate variability from these records by reducing measurement uncertainty through a combination of proxy data while retaining the temporal resolution needed to assess the timing and duration of variations in climate parameters."

*L54: 'the seasonal cycle is the most important cycle in Earth's climate.' This is a bold and rather subjective statement – I can think of many who might argue the carbon cycle is equally important. I'd suggest changing 'most' to 'one of the most' or omitting the statement all together.*

We understand this comment and will rephrase according to the reviewer's suggestion. The point we wanted to make is that the annual cycle is the most dominant cycle in climate variability when compared to cycles on other timescales (e.g. diurnal, orbital or tectonic timescales), but we realize that this is quite technical and distracts from the main point made in this sentence. The suggested rephrasing by the reviewer is the more elegant solution.

*L91: Colon missing after 'Optimization'*

We will add the colon here

*L120: Consider changing 'depth domain' to 'sampling domain' (here and in all subsequent references); ontogenetic trajectories aren't necessarily depths*

Good point, we will rephrase this here and throughout the manuscript.

*L121: Delete 'this' L139: There's an error in this equation (δ18Osw,freshwater is repeated twice). The mass balance should read: δ18Osw = f\* δ18Osw,freshwater + (1-f)\*δ18Osw,ocean*

Very good, we apologize for the mistake and thank the reviewer for pointing it out. We will rephase the second $\delta^{18}O$-term to $\delta^{18}O_{w,\,ocean}$. In addition, we will rephrase all mentions of $\delta^{18}O_{sw}$ ("$\delta^{18}O$ of the seawater") to $\delta^{18}O_w$ ("$\delta^{18}O$ of the water") to avoid the internal inconsistency of labelling $\delta^{18}O_w$-values with "sw, freshwater".

*L142: Space between the per mille symbol and VSMOW; this information is repeated on L145*

We will add the space between the symbol and "VSMOW" here and throughout the manuscript and remove the duplication in L145 as follows:

"…in the nearby Elbe and Weser rivers (see Ullmann et al., 2010)."

In addition, we will mention here that all $\delta^{18}O_{sw}$ values are given in ‰VSMOW throughout the manuscript to remove the mentions of VSMOW further in the text (see comment to L159)

*L157: Change 'is' to 'are'*

We will rephrase this accordingly

*L159, etc.: I'd suggest stating upfront that all references to seawater composition are in reference to VSMOW; the VSMOW reference after each composition is a bit clunky*

A good suggestion, and we will define $\delta^{18}O_{sw}$ values as given in per mille against VSMOW on first mention (L142)

*L217: Should be case 31 not 30, I believe*

Correct, we will scrutinize our references to case numbers throughout the manuscript in an attempt to catch similar mistakes.

*L272: Was the seasonal seawater composition range calculated based on the difference between warmest and coldest month (as described in the text) or between the most depleted and enriched δ18Osw seawater months? There are many examples where the SST and seawater cycles are out of phase (e.g., springtime snow melt flux driving a δ18Oseawater depletion extreme prior to the summer temperature extreme). The difference between peak summer and winter seawater composition is an important variable to constrain, but it does not necessarily always equate to the seasonal range in δ18Osw.*

This is a fair point which actually highlights a typographic error. We compare our results against the difference between the most depleted and enriched monthly $\delta^{18}O_{sw}$ values, which do not necessarily equate to summer and winter months (see calculations in **S5**). We will rephrase this accordingly. Our $\Delta_{47}$-based reconstruction approaches aim to disentangle the effects of $\delta^{18}O_{sw}$ and temperature on carbonate chemistry. The possibility of disentangling these two cycles, which are potentially out of phase, is one of the major advantages of $\Delta_{47}$-based seasonality reconstructions using microsampled carbonate records.

*L285: Is it fair to assume a normal marine in this environment? I suspect that this was done to illustrate a point and while I don't disagree that constraining mean annual (let alone seasonal variability) in δ18Osw in deep time adds huge uncertainty to conventional oxygen isotope interpretations, we can often make somewhat more realistic estimates of the (mean annual) value based on latitude and environment than just the global normal marine value.*

This is a fair point, but we chose to assume one $\delta^{18}O_{sw}$ value for all our $\delta^{18}O_c$-based reconstructions to mimic the process underlying most seasonality reconstructions based $\delta^{18}O_c$ measurements in real (fossil) carbonate records. We agree that in some fossil cases $\delta^{18}O_{sw}$ values are assigned with consideration of the depositional environment, but very often a fixed (non-seasonal) value is assigned based on general assumptions about the global ocean during the period under study. A good example is the often-cited ice-free marine $\delta^{18}O_{sw}$ value of -1‰VSMOW (after Shackleton, 1986). Alternatively, the chosen $\delta^{18}O_{sw}$ value is sometimes adjusted based on the $\delta^{18}O_c$-based temperature reconstructions, which risks circular reasoning. Our point is that, without independent evidence for the $\delta^{18}O_{sw}$ value, seasonality reconstructions based on $\delta^{18}O_c$ data rely on estimates of this value which may be very incorrect, especially in environments such as the one in the oyster study by Ullmann et al. (2010). We therefore think that the assumption of a $\delta^{18}O_{sw}$ value of 0‰ is not unrealistic for these cases. In addition, nearly all of our virtual datasets are based on a mean $\delta^{18}O_{sw}$ value of 0‰, so our assumption of this value as the "true" marine $\delta^{18}O_{sw}$ value in all cases might even underestimate the inaccuracy of $\delta^{18}O_c$-based reconstructions overall considering the large changes in global (let alone local or seasonal) $\delta^{18}O_{sw}$ that likely occurred in the geological past (see e.g. Veizer and Prokoph, 2015). To make this point more clearly and to motivate our decision, we will introduce the problem of uncertainty in the $\delta^{18}O_{sw}$ value in the Introduction.

*L294: Lower accuracy or higher offset*

We rephrase this to "lower accuracy". Note that we define accuracy as the agreement between reconstruction and "true" value, so within this definition high offset and low accuracy are synonymous. To avoid confusion, we will refrain from using "offset" to describe "accuracy" throughout the manuscript after giving our definition of accuracy (lines 98-99).

*L294: Change 'on' to 'of'*

We will rephrase this accordingly.

*LN14: Accuracy improves, offset decrease*

Correct, we will rephrase to: "Accuracy also increases with sampling resolution"

*LN14: Change 'samples year' to 'samples/year'*

We will rephrase this accordingly.

*LN128: Add a space between 'and' and 'Tagliavento'*

We will insert the space here.

*LN136: Higher precision or lower reproducible error*

Correct, this should read "high precision" and we will rephrase this accordingly.

*LN163: I'm unclear how 'event or spike' relate to the examples discussed here; I suggest omitting for clarity.*

Agreed, we will remove "event or spike".

*LN194: Change 'as' to 'at'*

We will rephrase this accordingly.

*LN231: Modify 'case of exceptions, in which' to 'cases in which' for clarity*

We will rephrase to: "In cases in which multi-annual variability in…"

*LN248: Change 'Cases 26' to 'Cases 2-6'*

We will rephrase this accordingly.

*LN380,382: Are the double hyphens (between 14–24 and 14–18) intentional?*

These are typographic errors and we will remove one of the hyphens.

*LN456: 'between 210 ppm' is not a range, I think a hyphen or a second number is missing*

Correct, this should read "2-10 ppm" and we will rephrase accordingly.

*Figures:*

*Figure 2: Consider adding heading labels for the blocks of virtual cases (e.g., a 'Sensitivity cases' header for 1-13, 'Natural cases' for 14-18, etc.). It will help focus the reader's eye to the differences between cases and will serve as a more useful reference throughout the results and discussion.*

An excellent suggestion, and we will add these headers.

*Figure 7: Color scheme is off in this figure – it looks like the smoothing and binning data may be reversed in the box plots.*

Correct, colours for these approaches have been accidentally reversed. We apologize for the mistake and will correct it in the revision whilst adapting the figure to the more accessible colour scheme.

*Figure 9: Please define RSD in the caption (the definition is there, but it is never directly linked to the acronym)*

We will add a definition to the caption.

*Supplement:*

*S12: Case 18 is missing (Natural case 5); numbering of all subsequent cases is off*

We thank the reviewer for pointing out this omission and will add data on case 18 to the table in the revision.

*References cited:*

*Keating-Bitonti, C.R., Ivany, L.C., Affek, H.P., Douglas, P. and Samson, S.D., 2011. Warm, not super-hot, temperatures in the early Eocene subtropics. Geology, 39(8), pp.771-774.*

**Anonymous Referee #2**

*In this manuscript De Winter et al present a complex study evaluating sampling and statistical methodologies aimed at constraining uncertainties in the application of stable isotope data to reconstruct past sea water temperature. Given the rapid rate at which new sclerochronological stable isotope records are being constructed, such an investigation is both timely and pertinent. Whilst this manuscript could provide an extremely useful outline of methodologies for future isotope studies, it is extremely difficult to follow. This is not helped by the nomenclature used and jumping between virtual and real isotope data and different methodologies without sufficient explanation for what these methodologies are and why they are being used. The readability of both text and figures therefore needs to be improved.*

We thank the reviewer for their comments and will pay special attention to the clarity of our terminology and explanation of our methodologies in the revised version.

*The main area of weakness in this manuscript is associated with the description of the methodology. The authors need to set out in far clearer terms the exact application of data in each of the three analytical processes and exactly which data is used. Given the readability and lack of clarity in the methodology, I would recommend major revisions be made to this manuscript prior to publication*

In response to this comment, we will clarify in greater detail how the three reconstruction approaches we test in this study are carried out and which data are used in each of these approaches. We will revise section 3.4 of our manuscript and Figure 4 (see comment below) to provide more detail on the workflow of these approaches. In addition, we have compiled the R scripts used to do the calculations into an R package, which is now available through the open-source online R database CRAN (https://cran.r-project.org/web/packages/seasonalclumped). In this package, the three $\Delta_{47}$-based approaches and the $\delta^{18}O_c$-based approach are explained in the package documentation and accompanied with a data on all virtual case studies used in this work for easy reference. We will refer to this package in the revised version of the manuscript, but we will also make sure that the explanations needed for the reader to understand and distinguish between the reconstruction approaches are present in the main manuscript text.

*Other comments:*

*Why was mean annual temperature used as a target? When isotope records are developed from species that have a known growth hiatus, for example during winter, it is unlikely that mean annual temperature would be targeted. An alternative target would be used, for example mean summer or a growing season mean instead. Targeting annual mean would make the record look worse than it actually is.*

We agree with the reviewer that many studies refer to growth temperatures in describing their reconstructions to acknowledge the fact that most carbonate records do not grow throughout the year. A similar case can be made for the use of seasonality defined as the difference between warmest/most $\delta^{18}O_{sw}$-depleted and coldest/most $\delta^{18}O_{sw}$-enriched month, which would also be underestimated in archives in which the monthly extremes are not recorded due to growth hiatuses. However, the goal of climate reconstruction in general is to arrive at (estimates of) climate parameters which can be compared with results from climate models or previous reconstructions independent from the growth season of the archive. We believe that the conversion from mean growth season temperature to mean annual temperature eventually needs to be made. In our comparison between reconstruction approaches, we wanted to consider any uncertainty or bias associated with variability in growth rate of the archive because it can bias real climate reconstructions (from fossil archives) as well. Therefore, we opted to compare our reconstruction results with climate parameters as they are officially defined (USGS; O'Donnell et al., 2012; see discussion on lines 175-184 on page 31-32, with our apologies for the discontinuity in the line numbering).

To further clarify this point, we briefly motivate our decision for choosing these benchmarks for comparison in the revised methods section where we give our definitions for accuracy and reproducibility.

*Pg 5 ln 89-91: The explanations for each of the methods are insufficient.*

We regret that the explanation of the approaches was not clear and will attempt to clarify them in the revised version (see reply to major comment).

*Line 195: "not exactly normally distributed" remove the word exactly.*

We will remove "exactly".

*Pg 29 ln 119: missing "to" between "equivalent the"*

Correct, we will insert "to" here.

*Pg 30 ln 142: 144 – "Carbonate records from suitable environments include, for example, the A. islandica bivalves from considerable depth (30-50m) in the open marine Northern Atlantic (e.g. Schöne et al., 2005, on which case 33 is based)." It would b inaccurate to assume that there is no variability in ï¸Ad'18Osw in NE Iceland. Variability in ï¸Ad'18Osw can also derive from changes in water mass that are bathing the shells during carbonate precipitation.*

This is a fair point, and we will rephrase this statement to reflect that we cannot exclude any seasonal variability in $\delta^{18}O_w$ in this setting:

"Carbonate records from environments with more stable $\delta^{18}O_w$ conditions include, for example, the *A. islandica* bivalves from considerable depth (30-50m) in the open marine Northern Atlantic (e.g. Schöne et al., 2005, on which case 33 is based), although even here variability in $\delta^{18}O_{sw}$ due to, for example, shifting influence of different bottom water masses cannot be fully excluded."

*Pg 30 ln 146: "the processes affecting δ18Osw on smaller scales" such as?*

We will elaborate a bit more on these processes by adding: ", such as local evaporation and freshwater influx from nearby rivers (e.g. Petersen et al., 2016)"

*Pg 30: 151-152: "While variability in δ18Osw compromises accurate δ18O-based seasonality reconstructions, the compilation in Fig. 3 shows that its influence on the δ18O records is too small to affect the shape of the record to such a degree that seasonality is fully obscured."*

It is not clear whether the reviewer would like to suggest any changes to this statement.

*Pg 36 ln 270 "While hiatuses encompassing half of the seasonal cycle are uncommon" I would not say that they are uncommon. There are many examples of sub-tropical to polar marine bivalve species which exhibit a cessation in growth during certain months of the year.*

We agree that severe slowing or completely cessation of growth during certain months of the year are common, but we retain that hiatuses masking half a year are very uncommon. The point we would like to make here is that in most natural cases and all tested cases except for case 18 archives record at least half of the seasonal cycle. This is important because most approaches to extract seasonal variability from carbonate archives become very inaccurate if more than half of the seasonal cycle is missing (see description in section 3.2). With less than half of the seasonal cycle present it becomes nearly impossible to recognize the season of growth and therefore to estimate seasonality or mean annual temperature in any meaningful way (see reply to general comment above). We will add a sentence after this statement to clarify this point and refer to the description in section 3.2.

*Pg 36 ln 280 missing "in" between "simulated the"*

We will insert "in" here.

*Sections 5.4.1 to 5.4.5 should be deleted. This is already a very long manuscript and this extra detail does not add anything to the focus of this manuscript.*

Both reviewers questioned the relevance of section 5.4 of this manuscript and we agree that this section makes the manuscript longer than necessary. However, we retain that a (short) discussion of the implications of this work beyond climate reconstructions is useful. Therefore, instead of deleting the entire section, we decided to summarize its content into one paragraph, following the suggestion of Referee #1.

*All figures: Colours used in Figures need to be changed to be accessible. The website https://colorbrewer2.org/ provides a useful free resource to check colour choices.*

We thank the reviewer for their suggestion and adapted all our figures according to the RdYlBu 4-colour scheme provided by Colorbrewer 2.0 (https://colorbrewer2.org/?type=diverging&scheme=RdYlBu&n=4) in order to make them more accessible to colour-blind readers.

*Fig 2: Whilst it is good to see a visual representation of each of the cases, the size of the panels makes it difficult to actually see what the purpose of each case is. It would be helped to add a schematic or table highlighting the purpose of each case.*

This is a good point, and echoes a comment by Referee #1. In reply to this comment, we will add headers to indicate the various categories of cases (e.g. "sensitivity tests", "natural cases", "Real SST/SSS data" etc.; see **S1**). In addition, we will try to include an abridge version of table **S1** to the main text to make this figure, and the difference between cases in general, easier to interpret by the reader.

*Fig 4A. Whilst a schematic would be a helpful figure to help the reader visualise what is a very complex methodology, the schematic presented in Fig 4A doesn't help. Currently this schematic does not help to clarify the methodology. I would suggest that the example provided in Fig 4V is removed and more space provided for panel A. This would provide space to add detail along with the arrows.*

This is a good suggestion and in the revised version we will limit Figure 4 to include only panel A. This would allow us to add more information next to the arrows detailing the workflow of creating virtual data and doing reconstructions. We would like to keep the example now included in panel B of Figure 4 in the main text and will include it as a separate figure.

*Fig 6. Axis text size needs to be increased.*

We agree that this figure is relatively dense and in response to a comment by Referee #1, we will try to represent the results of individual cases in a different way to make it easier to grasp the information. The present Figures 6 and 7 will then likely move to the supplement, where we will include them with larger text sizes.

*Fig 12: Change the colours on the plot, you can't tell the difference between the d18O and the binning results.*

We apologize for the fact that our choice of colours for the plots throughout the manuscript has compromised the clarity of our figures. As suggested, we used the Colorbrewer tool to pick more accessible colours and revise all our figures accordingly.

**References**

Keating-Bitonti, C. R., Ivany, L. C., Affek, H. P., Douglas, P. and Samson, S. D.: Warm, not super-hot, temperatures in the early Eocene subtropics, Geology, 39(8), 771–774, https://doi.org/10.1130/G32054.1, 2011.

Shackleton, N. J.: Paleogene stable isotope events, Palaeogeography, Palaeoclimatology, Palaeoecology, 57(1), 91–102, 1986.

Ullmann, C. V., Wiechert, U. and Korte, C.: Oxygen isotope fluctuations in a modern North Sea oyster (Crassostrea gigas) compared with annual variations in seawater temperature: Implications for palaeoclimate studies, Chemical Geology, 277(1), 160–166, 2010.

Veizer, J. and Prokoph, A.: Temperatures and oxygen isotopic composition of Phanerozoic oceans, Earth-Science Reviews, 146, 92–104, https://doi.org/10.1016/j.earscirev.2015.03.008, 2015.

---

## Author Comment (AC3) · 5 Feb 2021

Dear Alberto Reyes,

Thank you for guiding the review process. Please find our responses (AC1 and AC2) to the review comments in the interactive discussion. In addition to a general reply to the major concerns raised by the reviewers (AC1), we also provide a point-bt-point rebuttal in which we adress all reviewer comments (AC2).

Kind regards,

Niels J. de Winter

on behalf of the authors

---

## Author Response (AR1)

Dear Alberto Reyes, dear reviewers,

We would like to thank both reviewers for their constructive comments on our manuscript. We are confident that their suggestions helped us improve our manuscript and tried our best to implement them during our revision. Please find a detailed point-by-point rebuttal to the comments raised by the reviewers below. We hope that our replies to the reviewers' concerns and the revisions we propose will be adequate to merit publication of our manuscript in Climate of the Past. In the text below, we indicate review comments in **red** and our replies in **black**.

*Anonymous Referee #1*

*In 'Optimizing sampling strategies in high-resolution paleoclimate records', de Winter et al. propose four data treatment methods for constraining (paleo)climatological parameters from sequentially-sampled isotopic records and interrogate the accuracy and precision of each of these approaches using a combination of real and virtual datasets. Seasonally-resolved proxy data (e.g., δ18O and more recently ∆47) offer the potential to provide greater clarity into past climates. However, a number of factors complicate the interpretation of these archives. Conventional oxygen isotope thermometry requires an assumption regarding the isotopic composition of seawater, which, as the authors point out, is both spatially and seasonally variable even in the modern ocean. Clumped isotope thermometry is independent of seawater composition, circumventing this limitation, but comes with a much larger analytical uncertainty (made even larger by the small sample sizes required for sub-annual resolution) and thus the data can be noisy. By exploring how these and several other factors, such as sampling size and length of record, impact the accuracy and precision of recovered climatological information, the authors are able to develop a framework for determining the best sampling and statistical strategies to maximize the fidelity and minimize the uncertainty of these recovered parameters.*

*This study serves to significantly advance sclerochronologic research by providing critical insights into the uncertainties surrounding seasonal analyses and a quantitative, statistically-rooted means of extracting useful climate information from often noisy records. Overall, the paper is well-written, the experimental design well thought out, and the discussion thorough. The topic is pertinent and will be of interest to a broad paleoclimatological audience. I recommend only minor revisions, most of which pertain to improving the clarity and readability of the manuscript.*

*General comments:*

*(1) Based on Figs. 6, 7, and 12 it seems like the δ18O reconstructions aren't necessarily less accurate than the other approaches, but that the accuracy of the results are more variable. The approach accurately reconstructs MAT in Cases 1-14 and 19-29 and accurately reconstructs seasonal range in Cases 1-6, 19-24, and 30-33. Significant deviations occur only when the mean annual δ18Osw differs significantly from the assumed value and/or when there is a strong seasonality to δ18Osw (which, as you point out, is a realistic and often ignored scenario).*

*Given these findings, I'm curious why you have not attempted to combine the δ18O approach with another method to maximize both precision and accuracy of the results. If, for example, you were to use the binning or optimization of clumped isotope technique to constrain the seasonal seawater cycle, you could then apply those results to the higher-precision δ18O approach and alleviate the assumption of normalmarine/invariant seawater composition. For a real-world example of this approach see Keating-Bitonti et al. (2011), who use bulk summer and winter clumped samples to identify a summertime freshwater influx impacting their oxygen isotope values.*

This is a valid suggestion by the reviewer, and we agree that a combination of the low-precision clumped isotope method with (potentially) lower accuracy $\delta^{18}O_c$ measurements could yield an optimal compromise between the pros and cons of these methods. In fact, our "optimization" approach is based on this idea. However, the approach of combining bulk clumped isotope measurements and

high-resolution $\delta^{18}O_c$ measurements carried out by Keating-Botonti et al. (2011) has disadvantages, especially in cases where the $\delta^{18}O_w$ seasonality is out of phase with temperature seasonality (see comment on line 272 below). In addition, larger bulk measurements for clumped isotope analyses may significantly average temperature and $\delta^{18}O_w$ reconstructions of summer and winter seasons compared to microsampled transects. We will clarify the advantages and disadvantages of different methods for combining clumped and $\delta^{18}O_c$ measurements in seasonality studies by introducing the approach by Keating-Bitonti et al. (2011) and comparing it with our approaches in the discussion.

*(2) Though beyond the scope of the variables considered here, there are two additional complicating factors in real-world sclerochronologic data that may be worth mentioning at some point: (2a) Unequal sample time-averaging as a function of growth rate - in practical application (versus virtual subsampling), the number of days (or weeks or months) averaged into a single sample will vary; when the organism is growing quickly the sample will represent comparatively less time than when it is growing slowly, using the same diameter drill bit. If growth slowdowns/hiatuses, e.g., correspond with winter extremes, this results in not only fewer but also more time averaged (and thus dampened) estimates of winter temperature. (2b) Uncertainty in the seasonal phasing of SSTs in paleoclimate studies – even in instances were samples from a fossil specimen can be aligned along some reliable age model (be it via growth band counting or a statistical model), assigning those data points a calendar date is ambiguous at best. The timing of the date of maximum and minimum SST can vary considerably based on latitude, environment, and other local factors. This is a particularly important source of uncertainty to consider when binning the data by month – shifting the calendar date assumptions by, e.g., 15 days, would result in a whole new grouping of monthly data points and could significantly alter the results.*

These are very good suggestions and we incorporate these considerations in our revised discussion. Since both these complications relate to the timing of growth of sampled intervals, we will incorporate them into the section discussing the effects of uncertainties in the age model and growth rate variability (section 4.2.2 in the new version).

*(3) Regarding the figures, I really appreciate the consistent color theme throughout, but for accessibility purposes you may want to double check that the color scheme is colorblind safe. Additionally, there is a lot of really useful information that is not always easily extracted from the figures – I'd encourage you to think about creative ways of graphically presenting the information that will allow the reader to quickly glean the most important aspects. For example, you could include a heatmap showing how accuracy and precision vary by case and climatological parameters (similar to the colored conditional formatting of S12), which may be easier to interpret than the line plots.*

We appreciate the suggestion by the reviewer and paid special attention to the colour scheme of our figures, which was also a concern raised by Referee #2. We now use the RdYlBu 4-colour scheme provided by Colorbrewer 2.0 to make our figures colour-blind- and print-friendly (see https://colorbrewer2.org/?type=diverging&scheme=RdYlBu&n=4). To be honest, the multi-dimensional character of our results caused us some difficulties in coming up with a proper presentation method for aggregating all results in overview figures (Figures 6 and 7). We appreciate the reviewer's suggestion for revising our presentation method for these figures and agree that a colormap may be a more intuitive way to visualize this data. We have now implemented this idea and came up with a more reader-friendly visualization of the data now shown in figures 6 and 7. Hopefully, this makes these information-dense figures more straightforward to interpret. The original versions of the figures are retained in the supplement.

*(4) I'm not convinced that Section 5.4 (Implications for other sample size problems) C3 sufficiently contributes to the overall content of the manuscript to warrant inclusion. This section reads like a grant proposal rather than a discussion and could easily be condensed into a single one-paragraph section briefly outlining potential additional applications of the approach.*

Both reviewers expressed doubts about the added value of section 5.4 in its current form. We therefore decided to shorten this section into one paragraph, as suggested by Referee #1 rather than deleting the entire section following Referee #2. We opted to keep the more detailed discussion of the implications for future studies in the Supplement.

*(5) Prior to the final submission, the manuscript, supplement, and figures all require a thorough read through to ensure consistency of terminology, case numbers, and color scheme. For example, high precision/accuracy are at times conflated with high reproducibility error/offset in the text. Case numbers are not always consistent (both in the text and supplements), and it appears as if the colors have been switched in at least one figure. I've tried to point out examples of these in the line-by-line comments, but I'm sure that I've missed some. The supplemental files are hard to navigate - it would be helpful to include a 'Read Me' tab in applicable excel files defining all acronyms and abbreviations and descriptions of the information provided in each subsequent tab.*

This is a very valid concern, and we apologize for any mistakes that may have crept in our terminology or colour scheme in the initial version of the manuscript. For the revision, we revisited the manuscript, figures and supplements in detail and made sure all references and colours are consistent. In response to this comment and related comments by Referee #2, we also more clearly defined our use of terminology for describing reliability of reconstructions ("accuracy", "precision", "offset", etc.) and provide more detail in the supplements explaining the data tables. In addition, we have now compiled the R scripts of our reconstruction approaches into an R package. This package, including documentation of all scripts, is now freely available online through the CRAN database (https://cran.r-project.org/web/packages/seasonalclumped) to ensure easy access to the reader. We will cite the package in the revised version of the manuscript.

*Specific Comments.*

*\*NOTE: line numbers restart after L347; second set of line numbers indicated by LN*

Our apologies for the break in the line numbering, we will amend this in the revised version.

*Text:*

*L12: The term 'events' is a bit ambiguous here, particularly since the manuscript focuses on recovering climatological parameters (i.e., multi-year averages that smooth 'events')*

We acknowledge that this statement may be ambiguous because the length of an "event" may depend on its (broad) definition. To keep referring to the compromise between achieving accurate climate information by combining measurements on the one hand, and retaining a high temporal resolution in reconstructions on the other hand, we rephrase as follows:

"The challenge is to isolate meaningful information on climate variability from these records by reducing measurement uncertainty through a combination of proxy data while retaining the temporal resolution needed to assess the timing and duration of variations in climate parameters."

*L54: 'the seasonal cycle is the most important cycle in Earth's climate.' This is a bold and rather subjective statement – I can think of many who might argue the carbon cycle is equally important. I'd suggest changing 'most' to 'one of the most' or omitting the statement all together.*

We understand this comment and rephrased according to the reviewer's suggestion. The point we wanted to make is that the annual cycle is the most dominant cycle in climate variability when compared to cycles on other timescales (e.g. diurnal, orbital or tectonic timescales), but we realize that

this is quite technical and distracts from the main point made in this sentence. The suggested rephrasing by the reviewer is a more elegant solution.

*L91: Colon missing after 'Optimization'*

This section has been revised to introduce the different reconstruction approaches more clearly (section 2.1 in the new version). The colons are now redundant and have been removed.

*L120: Consider changing 'depth domain' to 'sampling domain' (here and in all subsequent references); ontogenetic trajectories aren't necessarily depths*

Good point, we rephrased this here and throughout the manuscript.

*L121: Delete 'this' L139: There's an error in this equation ($\delta 18 Osw, freshwater$ is repeated twice). The mass balance should read: $\delta 18 Osw = f * \delta 18 Osw, freshwater + (1-f) * \delta 18 Osw, ocean$*

Very good, we apologize for the mistake and thank the reviewer for pointing it out. We rephase the second $\delta^{18}O$-term to $\delta^{18}O_{w, ocean}$. In addition, we rephrased all mentions of $\delta^{18}O_w$ ("$\delta^{18}O$ of the seawater") to $\delta^{18}O_w$ ("$\delta^{18}O$ of the water") to avoid the internal inconsistency of labelling $\delta^{18}O_w$-values with "sw, freshwater".

*L142: Space between the per mille symbol and VSMOW; this information is repeated on L145*

We add the space between the symbol and "VSMOW" here and throughout the manuscript and remove the duplication in L145 as follows:

"…in the nearby Elbe and Weser rivers (see Ullmann et al., 2010)."

In addition, we mention here that all $\delta^{18}O_w$ values are given in ‰VSMOW throughout the manuscript to remove the mentions of VSMOW further in the text (see comment to L159)

*L157: Change 'is' to 'are'*

We rephrased this accordingly.

*L159, etc.: I'd suggest stating upfront that all references to seawater composition are in reference to VSMOW; the VSMOW reference after each composition is a bit clunky*

A good suggestion, and we define $\delta^{18}O_w$ values as given in per mille against VSMOW at the earliest opportunity (Lines 201-202 of the new version)

*L217: Should be case 31 not 30, I believe*

Correct, we scrutinized our references to case numbers throughout the manuscript to correct similar mistakes.

*L272: Was the seasonal seawater composition range calculated based on the difference between warmest and coldest month (as described in the text) or between the most depleted and enriched $\delta 18 Osw$ seawater months? There are many examples where the SST and seawater cycles are out of*

*phase (e.g., springtime snow melt flux driving a δ18Oseawater depletion extreme prior to the summer temperature extreme). The difference between peak summer and winter seawater composition is an important variable to constrain, but it does not necessarily always equate to the seasonal range in δ18Osw.*

This is a fair point which actually highlights a typographic error. We compare our results against the difference between the most depleted and enriched monthly $\delta^{18}O_w$ values, which do not necessarily equate to summer and winter months (see calculations in **S5**). We rephrase this accordingly. Our $\Delta_{47}$-based reconstruction approaches aim to disentangle the effects of $\delta^{18}O_w$ and temperature on carbonate chemistry. The possibility of disentangling these two cycles, which are potentially out of phase, is one of the major advantages of $\Delta_{47}$-based seasonality reconstructions using microsampled carbonate records (refer also to the discussion added in response to general comments 1 & 2).

*L285: Is it fair to assume a normal marine in this environment? I suspect that this was done to illustrate a point and while I don't disagree that constraining mean annual (let alone seasonal variability) in δ18Osw in deep time adds huge uncertainty to conventional oxygen isotope interpretations, we can often make somewhat more realistic estimates of the (mean annual) value based on latitude and environment than just the global normal marine value.*

This is a fair point, but we chose to assume one $\delta^{18}O_w$ value for all our $\delta^{18}O_c$-based reconstructions to mimic the process underlying most seasonality reconstructions based $\delta^{18}O_c$ measurements in real (fossil) carbonate records. We agree that in some fossil cases $\delta^{18}O_w$ values are assigned with consideration of the depositional environment, but very often a fixed (non-seasonal) value is assigned based on general assumptions about the global ocean during the period under study. A good example is the often-cited ice-free marine $\delta^{18}O_w$ value of -1‰VSMOW (after Shackleton, 1986). Alternatively, the chosen $\delta^{18}O_w$ value is sometimes adjusted based on the $\delta^{18}O_c$-based temperature reconstructions, which risks circular reasoning. Our point is that, without independent evidence for the $\delta^{18}O_w$ value, seasonality reconstructions based on $\delta^{18}O_c$ data rely on estimates of this value which may be very incorrect, especially in environments such as the one in the oyster study by Ullmann et al. (2010). We therefore think that the assumption of a $\delta^{18}O_w$ value of 0‰ is not unrealistic for these cases. In addition, nearly all of our virtual datasets are based on a mean $\delta^{18}O_w$ value of 0‰ (see discussion in section 4.1.1 in the revised manuscript), so our assumption of this value as the "true" marine $\delta^{18}O_w$ value in all cases might even underestimate the inaccuracy of $\delta^{18}O_c$-based reconstructions overall considering the large changes in global (let alone local or seasonal) $\delta^{18}O_w$ that likely occurred in the geological past (see e.g. Veizer and Prokoph, 2015). To make this point more clearly and to motivate our decision, we introduce the problem of uncertainty in the $\delta^{18}O_w$ value in the Introduction and discuss it in section 4.1.1.

*L294: Lower accuracy or higher offset*

We rephrase this to "lower accuracy". Note that we define accuracy as the agreement between reconstruction and "true" value, so within this definition high offset and low accuracy are synonymous. To avoid confusion, we will refrain from using "offset" to describe "accuracy" throughout the manuscript after giving our definition of accuracy (section 2.2).

*L294: Change 'on' to 'of'*

We rephrase this accordingly.

*LN14: Accuracy improves, offset decrease*

Correct, we rephrase to: "Accuracy also improves with sampling resolution"

*LN14: Change 'samples year' to 'samples/year'*

We rephrase this accordingly.

*LN128: Add a space between 'and' and 'Tagliavento'*

We insert the space.

*LN136: Higher precision or lower reproducible error*

Correct, this should read "high precision" and we rephrase this accordingly.

*LN163: I'm unclear how 'event or spike' relate to the examples discussed here; I suggest omitting for clarity.*

Agreed, we remove "event or spike".

*LN194: Change 'as' to 'at'*

We rephrase this accordingly.

*LN231: Modify 'case of exceptions, in which' to 'cases in which' for clarity*

We rephrase to: "In cases in which multi-annual variability in…"

*LN248: Change 'Cases 26' to 'Cases 2-6'*

We rephrase this accordingly.

*LN380,382: Are the double hyphens (between 14–24 and 14–18) intentional?*

These are typographic errors and we remove the second hyphen.

*LN456: 'between 210 ppm' is not a range, I think a hyphen or a second number is missing*

Correct, this should read "2-10 ppm" and we rephrase accordingly. Note that this section was now moved to the supplement, in the main text it is abbreviated to a summary paragraph.

*Figures:*

*Figure 2: Consider adding heading labels for the blocks of virtual cases (e.g., a 'Sensitivity cases' header for 1-13, 'Natural cases' for 14-18, etc.). It will help focus the reader's eye to the differences between cases and will serve as a more useful reference throughout the results and discussion.*

An excellent suggestion, and we added these headers in the revised figure.

*Figure 7: Color scheme is off in this figure – it looks like the smoothing and binning data may be reversed in the box plots.*

Correct, colours for these approaches have been accidentally reversed. We apologize for the mistake and corrected it in the revision whilst adapting the figure to the more accessible colour scheme.

*Figure 9: Please define RSD in the caption (the definition is there, but it is never directly linked to the acronym)*

We linked the abbreviation (RSD) to the definition in the revised caption.

*Supplement:*

*S12: Case 18 is missing (Natural case 5); numbering of all subsequent cases is off*

We thank the reviewer for pointing out this omission and added data on case 18 to the table in the revision.

*References cited:*

*Keating-Bitonti, C.R., Ivany, L.C., Affek, H.P., Douglas, P. and Samson, S.D., 2011. Warm, not super-hot, temperatures in the early Eocene subtropics. Geology, 39(8), pp.771-774.*

**Anonymous Referee #2**

*In this manuscript De Winter et al present a complex study evaluating sampling and statistical methodologies aimed at constraining uncertainties in the application of stable isotope data to reconstruct past sea water temperature. Given the rapid rate at which new sclerochronological stable isotope records are being constructed, such an investigation is both timely and pertinent. Whilst this manuscript could provide an extremely useful outline of methodologies for future isotope studies, it is extremely difficult to follow. This is not helped by the nomenclature used and jumping between virtual and real isotope data and different methodologies without sufficient explanation for what these methodologies are and why they are being used. The readability of both text and figures therefore needs to be improved.*

We thank the reviewer for their comments and paid special attention to the clarity of our terminology and explanation of our methodologies in the revised version.

*The main area of weakness in this manuscript is associated with the description of the methodology. The authors need to set out in far clearer terms the exact application of data in each of the three analytical processes and exactly which data is used. Given the readability and lack of clarity in the methodology, I would recommend major revisions be made to this manuscript prior to publication*

In response to this comment, we clarified in greater detail how the three reconstruction approaches we test in this study are carried out and which data are used in each of these approaches (section 2.1 in the revised manuscript). We also revised Figure 4 (see comment below) to provide more detail on the workflow of these approaches. In addition, we have compiled the R scripts used to do the calculations into an R package, which is now available through the open-source online R database CRAN (https://cran.r-project.org/web/packages/seasonalclumped). In this package, the three $\Delta_{47}$-based

approaches and the $\delta^{18}O_c$-based approach are explained in the package documentation and accompanied with a data on all virtual case studies used in this work for easy reference. We refer to this package in the revised version of the manuscript, but we also made sure that the explanations needed for the reader to understand and distinguish between the reconstruction approaches are present in the main manuscript text.

*Other comments:*

*Why was mean annual temperature used as a target? When isotope records are developed from species that have a known growth hiatus, for example during winter, it is unlikely that mean annual temperature would be targeted. An alternative target would be used, for example mean summer or a growing season mean instead. Targeting annual mean would make the record look worse than it actually is.*

We agree with the reviewer that many studies refer to growth temperatures in describing their reconstructions to acknowledge the fact that most carbonate records do not grow throughout the year. A similar case can be made for the use of seasonality defined as the difference between warmest/most $\delta^{18}O_w$-depleted and coldest/most $\delta^{18}O_w$-enriched month, which would also be underestimated in archives in which the monthly extremes are not recorded due to growth hiatuses. However, the goal of climate reconstruction in general is to arrive at (estimates of) climate parameters which can be compared with results from climate models or previous reconstructions independent from the growth season of the archive. We therefore believe that the conversion from mean growth season temperature to mean annual temperature eventually needs to be made. In our comparison between reconstruction approaches, we wanted to consider any uncertainty or bias associated with variability in growth rate of the archive because it can bias real climate reconstructions (from fossil archives) as well. Therefore, we opted to compare our reconstruction results with climate parameters as they are officially defined (USGS; O'Donnell and Ignizio, 2012; see discussion on lines 175-184 on page 31-32 of the previous manuscript version, with our apologies for the discontinuity in the line numbering).

To further clarify this point, we briefly motivate our decision for choosing these benchmarks for comparison in the revised methods section where we give our definitions for accuracy and reproducibility (new section 2.2).

*Pg 5 ln 89-91: The explanations for each of the methods are insufficient.*

We regret that the explanation of the approaches was not clear and attempted to clarify them in the revised version (see section 2.1 and Figures 4 and 5 of the revised manuscript).

*Line 195: "not exactly normally distributed" remove the word exactly.*

We remove "exactly".

*Pg 29 ln 119: missing "to" between "equivalent the"*

Correct, we inserted "to" here.

*Pg 30 ln 142: 144 – "Carbonate records from suitable environments include, for example, the A. islandica bivalves from considerable depth (30-50m) in the open marine Northern Atlantic (e.g. Schöne et al., 2005, on which case 33 is based)." It would b inaccurate to assume that there is no variability in ï¸Ad'18Osw in NE Iceland. Variability in ï¸Ad'18Osw can also derive from changes in water mass that are bathing the shells during carbonate precipitation.*

This is a fair point, and we rephrased this statement to reflect that we cannot exclude all seasonal variability in $\delta^{18}O_w$ in this setting:

"Carbonate records from environments with more stable $\delta^{18}O_w$ conditions include, for example, the *A. islandica* bivalves from considerable depth (30-50m) in the open marine Northern Atlantic (e.g. Schöne et al., 2005, on which case 33 is based), although even here variability in $\delta^{18}O_w$ due to, for example, shifting influence of different bottom water masses cannot be fully excluded."

*Pg 30 ln 146: "the processes affecting δ18Osw on smaller scales" such as?*

We elaborate a bit more on these processes by adding: ", such as local evaporation and freshwater influx from nearby rivers (e.g. Surge et al., 2001; Petersen et al., 2016)"

*Pg 30: 151-152: "While variability in δ18Osw compromises accurate δ18O-based seasonality reconstructions, the compilation in Fig. 3 shows that its influence on the δ18O records is too small to affect the shape of the record to such a degree that seasonality is fully obscured."*

It is not clear whether the reviewer would like to suggest any changes to this statement.

*Pg 36 ln 270 "While hiatuses encompassing half of the seasonal cycle are uncommon" I would not say that they are uncommon. There are many examples of sub-tropical to polar marine bivalve species which exhibit a cessation in growth during certain months of the year.*

We agree that severe slowing or complete cessations of growth during certain months of the year are common, but we retain that hiatuses masking half a year are very uncommon. The point we would like to make here is that in most natural cases and all tested cases except for case 18 archives record at least half of the seasonal cycle. This is important because most approaches to extract seasonal variability from carbonate archives become very inaccurate if more than half of the seasonal cycle is missing (see discussion in 4.2.2 of the revised manuscript). With less than half of the seasonal cycle present it becomes nearly impossible to recognize the season of growth and therefore to estimate seasonality or mean annual temperature in a meaningful way (see reply to general comment above). We addressed this point in the revised discussion (section 4.2.2).

*Pg 36 ln 280 missing "in" between "simulated the"*

We will insert "in" here.

*Sections 5.4.1 to 5.4.5 should be deleted. This is already a very long manuscript and this extra detail does not add anything to the focus of this manuscript.*

Both reviewers questioned the relevance of section 5.4 of this manuscript and we agree that this section makes the manuscript longer than necessary. However, we retain that a (short) discussion of the implications of this work beyond climate reconstructions is useful. Therefore, instead of deleting the entire section, we decided to summarize its content into one paragraph, following the suggestion of Referee #1. The full-length version of this discussion chapter is kept in the supplement.

*All figures: Colours used in Figures need to be changed to be accessible. The website https://colorbrewer2.org/ provides a useful free resource to check colour choices.*

We thank the reviewer for their suggestion and adapted all our figures according to the RdYlBu 4-colour scheme provided by Colorbrewer 2.0

(https://colorbrewer2.org/?type=diverging&scheme=RdYlBu&n=4) to make them more accessible to colour-blind readers.

*Fig 2: Whilst it is good to see a visual representation of each of the cases, the size of the panels makes it difficult to actually see what the purpose of each case is. It would be helped to add a schematic or table highlighting the purpose of each case.*

This is a good point which echoes a comment by Referee #1. In reply to this comment, we added headers to indicate the various categories of cases (e.g. "sensitivity tests", "natural cases", "Real SST/SSS data" etc.). In addition, we will included an abridged version of table **S1** to the main text (Table 1) to make this figure, and the difference between cases in general, easier to interpret by the reader.

*Fig 4A. Whilst a schematic would be a helpful figure to help the reader visualise what is a very complex methodology, the schematic presented in Fig 4A doesn't help. Currently this schematic does not help to clarify the methodology. I would suggest that the example provided in Fig 4V is removed and more space provided for panel A. This would provide space to add detail along with the arrows.*

This is a good suggestion and in the revised version we limit Figure 4 to include what was previously labelled as panel A. This allowed us to add more text detailing the workflow of creating virtual data and reconstructing SST and $\delta^{18}O_w$. We kept the example previously included in panel B of Figure 4 in the main text and will include it as a separate figure (Figure 5).

*Fig 6. Axis text size needs to be increased.*

We agree that this figure is relatively dense and in response to a comment by Referee #1, we represented the results of individual cases in a heatmap fitting the new color sceme to make it easier to grasp the information. The previous Figures 6 and 7 have moved to the supplement.

*Fig 12: Change the colours on the plot, you can't tell the difference between the d18O and the binning results.*

We apologize for the fact that our choice of colours for the plots throughout the manuscript has compromised the clarity of our figures. As suggested, we used the Colorbrewer tool to pick more accessible colours and revise all our figures accordingly.

**References**

Keating-Bitonti, C. R., Ivany, L. C., Affek, H. P., Douglas, P. and Samson, S. D.: Warm, not super-hot, temperatures in the early Eocene subtropics, Geology, 39(8), 771–774, https://doi.org/10.1130/G32054.1, 2011.

O'Donnell, M. S. and Ignizio, D. A.: Bioclimatic predictors for supporting ecological applications in the conterminous United States, US Geological Survey Data Series, 691(10), 2012.

Shackleton, N. J.: Paleogene stable isotope events, Palaeogeography, Palaeoclimatology, Palaeoecology, 57(1), 91–102, 1986.

Ullmann, C. V., Wiechert, U. and Korte, C.: Oxygen isotope fluctuations in a modern North Sea oyster (Crassostrea gigas) compared with annual variations in seawater temperature: Implications for palaeoclimate studies, Chemical Geology, 277(1), 160–166, 2010.

Veizer, J. and Prokoph, A.: Temperatures and oxygen isotopic composition of Phanerozoic oceans, Earth-Science Reviews, 146, 92–104, https://doi.org/10.1016/j.earscirev.2015.03.008, 2015.

---

## Referee Report (RR1)

[referee-annotated manuscript omitted]

---

## Editor Decision (ED1)

**Optimizing sampling strategies in high-resolution paleoclimate records**

Niels J. de Winter[1,2] *, Tobias Agterhuis[1], Martin Ziegler[1]

[1]Department of Earth Sciences, Utrecht University, Princetonlaan 8a, 3584 CB Utrecht, the Netherlands

[2]AMGC research group, Vrije Universiteit Brussel, Pleinlaan 2, 1050 Brussels, Belgium

Correspondence to: Niels J. de Winter (n.j.dewinter@uu.nl)

**Abstract**

The aim of paleoclimate studies to resolve climate variability from noisy proxy records can in essence be reduced to a statistical problem. The challenge is to extract meaningful information about climate variability from these records by reducing measurement uncertainty through a combination of proxy data while retaining the temporal resolution needed to assess the timing and duration of variations in climate parameters. In this study, we explore the limits of this compromise by testing different methods for combining proxy data (smoothing, binning and sample size optimization) on a particularly challenging paleoclimate problem: resolving seasonal variability in stable isotope records. We test and evaluate the effects of changes in the seasonal temperature and the hydrological cycle as well as changes in accretion rate of the archive and parameters such as sampling resolution and age model uncertainty on the reliability of seasonality reconstructions based on clumped and oxygen isotope analyses in 33 real and virtual datasets. Our results show that strategic combinations of clumped isotope analyses can significantly improve the accuracy of seasonality reconstructions compared to conventional stable oxygen isotope analyses, especially in settings where the isotopic composition of the water is poorly constrained.

Smoothing data using a moving average often leads to an apparent dampening of the seasonal cycle, significantly reducing the accuracy of reconstructions. A statistical sample size optimization protocol yields more precise results than smoothing. However, the most accurate results are obtained through monthly binning of proxy data, especially in cases where growth rate or water composition cycles obscure the seasonal temperature cycle. Our analysis of a wide range of natural situations reveals that the effect of temperature seasonality on oxygen isotope records almost invariably exceeds that of changes in water composition. Thus, in most cases, oxygen isotope records allow reliable identification of growth seasonality as a basis for age modelling in absence of independent chronological markers in the record. These specific findings allow us to formulate general recommendations for sampling and combining data in paleoclimate research and have implications beyond the reconstruction of seasonality. We briefly discuss the implications of our results for solving common problems in paleoclimatology and stratigraphy.

**1. Introduction**

*[Handwritten margin note: is ʃt worth adding some text here to provide context on the issue of dlse model uncertainty, which you tackle later?]*

[revised manuscript text omitted]

---

## Author Response (AR2)

Dear Alberto Reyes and Andrew Johnson,

Thank you both for your comments and suggestions, which have improved our manuscript in this second review round. We are glad to read that you both agree with our revisions in the previous review round and are happy to follow your suggestions for the final minor revisions before the manuscript is published in Climate of the Past.

Please find below our point-by-point rebuttal (in **black**) to the last round of comments (in **red**), as they were provided in the online submission system. In addition to the comments below, both of you provided (digitally or manually) annotated PDFs with in-line comments. Since these comments were quite minor and specific and since we followed all the suggestions in these PDFs, we have decided to summarize the major points in these annotated files into our line-by-line rebuttal here.

Comments to the Author:

Dear Dr. de Winter,

Your manuscript has now been assessed by a third independent reviewer. Their review is positive and points to the importance of this comprehensive manuscript. You'll see that there are still some suggestions for increased clarity through some minor revision. I concur with this assessment, and add my own comments for improved clarity and some potential organizational changes in the attached annotated PDF. Accordingly, I'm recommending that the manuscript be accepted for publication pending minor revisions, to address my comments and those of the third reviewer.

Sincerely,

Alberto Reyes

We would like to specifically thank the editor for taking the time to provide detailed feedback on our manuscript in both review rounds. We implement his comments provided in the annotated PDF in an attempt to improve the structure of our manuscript and clarity of our formulation where necessary.

In reply to the editor's in-line comments, we modified the order in which our test cases are presented in the Methods section. We decided to keep the order of presentation in the Results because reversing the order would compromise the structure of presentation there.

We also followed the editor's suggestion to present our three types of cases in terms of measured vs. virtual environmental and proxy data to further clarify the difference between the three categories.

Throughout the manuscript, we revised all references to "precision error". We agree that this formulation is confusing and chose to only refer to low or high precision or directly to precision standard deviations if the reference is to the numeric result specifically. We also removed a few instances where "reproducibility" was used and rephrased this to "precision" to be consistent on terminology.

The introduction now includes a few sentences introducing the issue of age modelling within seasonally resolved records, as suggested.

Comments by Andrew Johnson

This is a thorough analysis of the factors affecting the accuracy and precision of environmental information determined from stable isotope data. It shows how their effects may be mitigated by combining/aggregating data in various ways and concludes with recommendations for the best approaches to adopt in particular circumstances and to achieve particular ends.

This is a complex paper and the previous version was evidently difficult to follow (see the comments of Reviewer 2 especially). The intended changes signalled in the authors' response to the comments of reviewers 1 and 2 have been implemented such that all the specific points have been addressed and clarity improved. The inclusion of the heatmaps (Figs. 6 and 7) suggested by Reviewer 1 is a good example of a very positive response to comments on clarity. Nevertheless, the text is still difficult to follow in places due to insufficient explanation, small errors, lack of reference to figure parts, etc. I have made comments and suggestions at the relevant points (highlighted text – sometimes very short sections) in the attached annotated version of the manuscript. These should all be considered and changes implemented as appropriate.

We thank the reviewer for his detailed comments in the annotated PDF and did our best to implement these in our revised manuscript in an attempt to make our discussion easier to follow.

In identifying statistical 'solutions' the paper points up data 'problems'. The latter are being reduced by analytical improvements (e.g. the sample size needed for $\Delta 47$ has reduced in recent years and the analytical precision has, I think, increased) and, as the authors briefly indicate (e.g. the larger number of samples obtainable from the early, rapid stage of shell growth), data quality can be improved by appropriate targeting. Given the confounding effect of $\delta 18O_w$ variation on estimates of temperature from $\delta 18O_c$, which the authors amply demonstrate, I really think they should take the opportunity to recommend that $\delta 18O_c$-based estimates of marine temperature are not conducted on euryhaline organisms. There are plenty of stenohaline marine organisms (including some that grow fast and continuously in early ontogeny) that are far better targets than the euryhaline oysters investigated in many previous works!

We agree with the reviewer that the effect of $\delta^{18}O_w$ variability (especially on the seasonal scale) on $\delta^{18}O_c$-based reconstructions is one of the main issues our study addresses. Indeed, the growth locality of the carbonate producer (e.g. estuarine vs. full marine conditions) strongly controls the size of this problem. However, while euryhaline organisms (such as oysters) generally grow in environments with more salinity (and thus $\delta^{18}O_w$) variability than stenohaline species, this does not, in our opinion, disqualify $\delta^{18}O_c$-based reconstructions from euryhaline species, nor does it entail that $\delta^{18}O_c$-based reconstructions from stenohaline organisms are not affected by this $\delta^{18}O_w$ variability. In reply to this comment, we discuss the difference between euryhaline and stenohaline organisms in a more nuanced way, stating that the problem of $\delta^{18}O_w$ variability plays a larger role in euryhaline organisms while the effect of $\delta^{18}O_w$ variability should be considered in every $\delta^{18}O_c$-based reconstruction.

Identifying suitable targets, referring to ongoing analytical improvements, and mentioning the possibility of constraining $\delta 18Ow$ by modelling (see comment at line 505), would enable a slightly more positive/optimistic tone in the conclusions.

This is a good comment, and we use the reviewer's comment on line 505 as an opportunity to mention some of these techniques as recommendations for constraining $\delta^{18}O_w$ variability in $\delta^{18}O_c$-based reconstructions (lines 582-589 of the revised manuscript). In the end, our recommendation

remains that actual independent temperature or $\delta^{18}O_w$ reconstructions (e.g. using the presented $\Delta_{47}$-based approach) should be favoured over these methods of approximating $\delta^{18}O_w$.